# Convergence of Clipped SGD on Convex $(L_0, L_1)$-Smooth Functions

**Ofir Gaash**[*]  **Kfir Yehuda Levy**[†]  **Yair Carmon**[*]

## Abstract

We study stochastic gradient descent (SGD) with gradient clipping on convex functions under a generalized smoothness assumption called $(L_0, L_1)$-smoothness. Using gradient clipping, we establish a high probability convergence rate that matches the SGD rate in the $L$ smooth case up to polylogarithmic factors and additive terms. We also propose a variation of adaptive SGD with gradient clipping, which achieves the same guarantee. We perform empirical experiments to examine our theory and algorithmic choices.

## 1 Introduction

Gradient clipping is a common method for stabilizing neural network training. Despite its wide use, little thought is given to the choice of the clipping *threshold*, with many works fixing it at 1 without attempting to tune it [3, 5, 34, 35, 2, 20, 21]. In pursuit of a theory-driven threshold choice, recent research tries to better understand the benefits of gradient clipping.

Experiments suggest that clipping is effective in situations where small changes in input can lead to significant variations in gradient norms [44, 43]. This observation has led Zhang et al. [44] to formally define this behavior as the following "relaxed" smoothness property.

**Definition 1.** A twice-differentiable function $f : \mathbb{R}^d \to \mathbb{R}$ is $(L_0, L_1)$-smooth if for every $x \in \mathbb{R}^d$ it holds that $\|\nabla^2 f(x)\| \le L_0 + L_1 \|\nabla f(x)\|$.

In words, $(L_0, L_1)$-smoothness allows the Hessian norm to increase linearly in the gradient norm. This is opposed to the traditional smoothness property, which says that the Hessian norm is bounded by a constant, coinciding with the new definition for $L_1 = 0$. Building on this definition, Zhang et al. [44] show instances where clipped SGD outperforms standard SGD. Specifically, they denote $M = \sup\{\|\nabla f(x)\| \mid f(x) < f(x_0)\}$, and show that the complexity of SGD with a fixed stepsize is larger than the complexity of SGD with gradient clipping by a factor of $L_1 M$ (assuming both algorithms are initialized at $x_0$). This factor can be very large: in particular, it may be exponential in $L_1 R_0$, where $R_0$ is the initial distance from the optimum (see Appendix G).

The connection between $(L_0, L_1)$-smoothness and gradient clipping has led researchers to attempt to characterize the complexity of $(L_0, L_1)$-smooth optimization, mainly in terms of the dependence on $L_1$. The non-convex setting is well understood: the state-of-the-art rate of gradient norm convergence comprises of the rate for $L_0$-smooth functions and additional low-order terms that depend on $L_1$. This holds for both clipped GD [38] and clipped SGD [37] (see Section 2 for details).

**Motivation for studying the convex setting.** Despite being originally motivated by the behavior of (non-convex) neural networks, generalized smoothness is also compelling to study in the convex setting. There are several instances where convex analysis closely aligns with empirical behavior; among them are AdaGrad-based algorithms and the momentum technique, in which theory preceded

---

[*]Tel Aviv University, `ofirgaash@mail.tau.ac.il` and `ycarmon@gmail.com`.

[†]Technion, `kfiryehud@gmail.com`.

39th Conference on Neural Information Processing Systems (NeurIPS 2025).

practice [29, 7, 32]. Furthermore, a body of work show that neural networks have convex-like behavior [14, 46, 22]. From a theoretical perspective, it allows to examine whether the pattern observed for gradient norm convergence—an $L_0$-smooth rate with low-order $L_1$-dependent terms—also holds for *optimality gap* convergence.

These considerations have motivated recent work to study the convex, $(L_0, L_1)$-smooth setting [15, 18, 11, 38, 37, 24]. For clipped GD, [11, 38, 37, 24] prove an optimality gap convergence rate following the aforementioned pattern. However, prior work does not provide a corresponding result in the stochastic setting (we discuss a concurrent work by Lobanov and Gasnikov [23] in Section 2). The difficulty of the stochastic, convex regime stems from the fact that clipping biases stochastic gradients. Bias is arguably more challenging in the convex setting than in the non-convex setting: the former intimately relies on stochastic gradients being unbiased, while for the latter it suffices to average enough stochastic gradient such that the noise drops below the required degree of stationarity [15]. This is potentially the reason previous studies of the convex setting [18, 15] proved convergence only in the deterministic regime.

**Our contribution.** In this work, we analyze gradient clipping in the convex, stochastic, $(L_0, L_1)$-smooth setting with light-tailed noise. Our contributions are as follows.

- We show that the pattern of an $L_0$-smooth convergence rate with $L_1$-dependent low-order terms extends to the stochastic convex setting. For clipped SGD with $\sigma$-sub-Gaussian gradient noise, we prove a sub-optimality bound of $O\left(\frac{\log(T/\delta)L_0R_0^2}{T} + \frac{\log^2(T/\delta)\sigma R_0}{\sqrt{T}}\right)$ that holds with probability at least $1 - \delta$ for $T = \Omega(\log\left(\frac{T}{\delta}\right)L_1^2R_0^2)$. That is, we bound the stochastic gradient query complexity of achieving optimality gap $\epsilon$ with probability at least $1 - \delta$ by $\widetilde{O}\left(\frac{L_0R^2}{\epsilon} + \frac{\sigma^2R_0^2}{\epsilon^2} + (L_1R_0)^2\right)$, where $\widetilde{O}(\cdot)$ hides factors poly-logarithmic in $\frac{1}{\delta\epsilon}$. This matches the best known bounds for (fixed step-size) SGD in the $L_0$-smooth case, which are $\Omega\left(\frac{L_0R^2}{\epsilon} + \frac{\sigma^2R_0^2}{\epsilon^2}\right)$ [16]. Our bound depends on $L_1$ only through an additive term, with no implicit dependence on $\exp(L_1R_0)$.

- We show the same complexity bound for two variations of SGD: adaptive SGD [25] with clipping, and SGD with a variable stepsize which we refer to as "implicit clipping."

- We show that, to achieve the bounds above, precise knowledge of the parameter $L_1$ is not required: simply replacing $L_1$ by the conservative choice $\widetilde{O}(T^{1/2}/R_0)$ suffices. This result does not appear in papers studying the deterministic regime.

- We perform numerical experiments to test the impact of gradient clipping for SGD on convex, $(L_0, L_1)$-smooth functions, and assess the empirical effect of some of our algorithmic choices.

To address the bias of stochastic gradients that is introduced by clipping, we employ a *double sampling* technique that uses two independent stochastic gradient samples in each update: one to estimate the direction of the update, and another to estimate its magnitude. This technique is necessary in order to apply some of the probabilistic tools we use, which assume sequences of unbiased random variables, but our empirical analysis suggests it might not be helpful in practice. We elaborate on this technique in the analysis as well as in the experiments.

The paper organization is as follows. In Section 2 we survey related work. In Section 3 we outline our algorithmic framework and prove our theoretic results. In Section 4 we describe our experiments and discuss their results. In Section 5 we provide a short conclusion.

## 2 Related work

**Generalized smoothness definitions.** Zhang et al. [44] first introduced the concept of $(L_0, L_1)$-smoothness as in Definition 1. Zhang et al. [43], Gorbunov et al. [11], Vankov et al. [38] provide useful equivalent definitions. Li et al. [18] introduce an even more general notion of smoothness: given a non-decreasing continuous function $\ell$, they define $\ell$-smoothness as the property $\|\nabla^2 f(x)\| \leq \ell(\|\nabla f(x)\|)$. Yu et al. [42] extend $\ell$-smoothness to $\ell*$-smoothness by allowing the choice of a non-euclidean norm. In this work, we focus on $(L_0, L_1)$-smoothness.

**Algorithms for $(L_0, L_1)$-smooth optimization.** Most prior work [44, 43, 30, 15, 11, 38, 24, 37] consider GD/SGD with gradient clipping, where the stepsize is of the form $\eta' \min\left\{1, \frac{c}{\|g\|}\right\}$ or $\eta' \frac{c}{\|g\|+c}$, where $g$ is a (possibly stochastic) gradient. Zhang et al. [44], Vankov et al. [38] show that these two forms are closely related. While we focus on clipping methods, other methods are also analyzed under $(L_0, L_1)$-smoothness, such as normalized stepsizes [45, 4, 41], Polyak stepsizes [33, 11, 38], coordinate descent methods [24], adaptive SGD [8, 39, 13] and Adam [17, 12, 40].

**Gradient clipping for $(L_0, L_1)$-smooth functions.** The non-convex regime was the first to be explored. Zhang et al. [44] provide the first theoretical demonstration of the advantage of clipped GD, and Koloskova et al. [15], Vankov et al. [38], Tyurin [37] subsequently improve the bounds to $O\left(\sqrt{L_0\Delta/T} + L_1\Delta/T\right)$. Clipped SGD is considered under several noise assumptions. Zhang et al. [44, 43] consider $\sigma$-bounded gradient noise with probability 1. Zhang et al. [44] prove a rate of $O\left((\Delta')^2/T^{1/4} + (L_0 + L_1\sigma)\Delta'/T^{1/2} + L_1\Delta'/T\right)$, where $\Delta' = \Delta + (L_0 + L_1\sigma)\sigma^2 + \sigma L_0^2/L_1$. Zhang et al. [43] prove a rate of $O\left(L_0\sigma^2\Delta/T^{1/4}\right)$ for $T = \Omega\left(L_1^4/L_0^3\right)$. While the former has no conditions on $T$, the latter has better dependency on the problem parameters. Li et al. [18], Koloskova et al. [15] consider $\sigma$-bounded gradient noise variance. Li et al. [18] prove a rate that has a dependency on the initial gradient norm, and Koloskova et al. [15] show an unavoidable bias term when considering all clipping thresholds at once. Tyurin [37] consider light-tailed noise, and using a batch size of $O\left(\sigma^2 T^2\right)$, obtain a rate of $O\left(L_1\Delta/T + \sqrt{L_0\Delta/T}\right)$.

The convex regime recently received much attention. Koloskova et al. [15], Li et al. [18] prove a convergence rate for clipped GD where the dominating term is $O\left((L_0 + ML_1)R_0^2/T\right)$, where $M$ is the maximal gradient norm among the iterates. Yu et al. [42] extend the result of Li et al. [18] to Mirror Descent. As discussed in the introduction, the term $M$ may be exponential in $L_1R_0$. Gorbunov et al. [11], Vankov et al. [38] independently prove a convergence rate of $O\left(L_0R_0^2/T\right)$ with additive factors of $L_1^2R_0^2$ and $\min\left\{L_1^2R_0^2, L_1R_0\log(\Delta T)\right\}$, respectively. Lobanov et al. [24] show that in some initial phase of the algorithm, clipped GD enjoys linear convergence. Gorbunov et al. [11], Vankov et al. [38] both propose acceleration methods; The former has an exponential dependency on $R_0$, and the latter requires to solve a one-dimensional optimization problem in each iteration.

In the stochastic convex case, Gorbunov et al. [11] consider finite-sum functions, with an additional assumption that all the functions share a common minimizer. They show a convergence rate of $O\left(L_0R_0^2/T\right)$ in expectation for $T = \Omega\left(nL_1^2R_0^2\right)$, where $n$ is the number of functions. In concurrent and independent work, Lobanov and Gasnikov [23] present a general framework and study both first- and zero-order methods. Assuming bounded noise variance, they obtain bounds for arbitrary clipping thresholds, and prove linear convergence in some special cases. However, their rate of convergence depends on $M$ and $\Delta$, both of which may be exponential in $L_1R_0$. Another concurrent and independent work [42] studies Stochastic Mirror Descent on $\ell*$-smooth functions. Under a noise variance assumption that generalizes over affine noise, they prove a high-probability, anytime convergence bound. Their rate, too, has an implicit exponential dependence on $L_1R_0$.

To the best of our knowledge, our work provides the first rate of convergence for stochastic, convex, $(L_0, L_1)$-smooth optimization without exponential dependence on $L_1R_0$, and with a leading-order term independent of $L_1$. For the case of $\sigma = 0$, our results match the state-of-the-art rate from the deterministic regime up to a logarithmic factor.[3] Our results hold with high probability, and the dependence on $1/\delta$ is poly-logarithmic.

**Adaptive SGD for $(L_0, L_1)$-smooth functions.** Faw et al. [8], Wang et al. [39] consider adaptive SGD for non-convex functions with an affine variance assumption. In the context of bounded variance, their rates translate to $\frac{(L_1\Delta+\sigma)^2}{\delta^2 T} + \frac{\sigma(L_1\Delta+\sigma)}{\delta^2\sqrt{T}}$. In a similar setting, Hong and Lin [12] prove a rate that is logarithmic in $1/\delta$, but polynomial in the dimension.

Our work on adaptive methods has several differences from the above. First, we analyze a *clipped* variation of adaptive SGD. Second, we consider convex functions and assume a stronger noise assumption. Third, our rate simultaneously has a poly-logarithmic dependence on $1/\delta$, is *not*

---

[3]Note that the work of Lobanov et al. [24] is not directly comparable to ours: in the regime $L_0 > 0$, their linear convergence does not hold asymptotically, but rather for some initial phase of the run.

dimension-dependent, and has a weaker dependence on $L_1$. Lastly, as for clipped SGD, we match the state-of-the-art result from the deterministic setting up to a logarithmic factor.

## 3 Analysis

**Notation.** Throughout the paper, $\|\cdot\|$ is the Euclidean norm, $\langle\cdot,\cdot\rangle$ is the Euclidean dot product, $\text{Proj}_{\mathcal{X}}(\cdot)$ is the Euclidean projection onto the set $\mathcal{X}$ and $\mathbb{B}(x,r)$ is the Euclidean ball of radius $r$ centered at $x$. We denote $\log_+(\cdot) := 2 + \log(\cdot)$, $R_t := \|x_t - x^\star\|$ and $\Delta_t := f(x_t) - f(x^\star)$.

**Problem setting.** We consider the optimization problem

$$\underset{x \in \mathbb{R}^d}{\text{minimize}} \, f(x)$$

where the function $f$ satisfies the following.

**Assumption 1.** The function $f : \mathbb{R}^d \to \mathbb{R}$ is convex, and $f$ attains a minimum at some $x^\star \in \mathbb{R}^d$ with distance at most $R$ from the initialization.

**Assumption 2.** The function $f$ is twice-differentiable[4] and $(L_0, L_1)$-smooth (see Definition 1).

We remark that the distance bound $R$ is only required for our analysis of Clipped Adaptive SGD. Specifically, Theorem 1 does not require it (and does not require the corresponding projection operator present in Algorithm 1).

We assume access to a stochastic first-order oracle $\mathcal{G}$ that satisfies $\mathbb{E}[\mathcal{G}(x) \mid x] = \nabla f(x)$ for any $x \in \mathbb{R}^d$. We consider two different noise assumptions.

**Assumption 3** (Bounded noise). The oracle $\mathcal{G}$ satisfies

$$\mathbb{P}(\|\mathcal{G}(x) - \nabla f(x)\|^2 \leq \sigma^2) = 1.$$

**Assumption 4** (Sub-Gaussian noise). The oracle $\mathcal{G}$ satisfies

$$\mathbb{E}\big[\exp\big(\|\mathcal{G}(x) - \nabla f(x)\|^2 / \sigma^2\big)\big] \leq \exp(1).$$

**Notes on our assumptions.** In Assumption 1, the existence of a minimum is required for a technical step in our main high-probability argument (Lemma 8). The bound $R$ allows us to analyze AdaGrad-like algorithms without explicitly constraining the objective's domain, thereby letting us use a fundamental lemma on $(L_0, L_1)$-smoothness (Lemma 1). Assumption 4 is a standard light-tail noise enabling high-probability bounds [44, 43, 37]. We conduct most of our analysis under a stronger bounded noise assumption (Assumption 3) and then use a reduction [1] to lift our result to hold under the weaker Assumption 4. It is sometimes possible to prove high probability bounds under even more relaxed moment-bound assumptions [6, 9, 10, 26, 27, 31]; doing so in our setting is an interesting topic for future work.

**Algorithms.** Our proposed methods are applications of Algorithm 1, which has three parameters:

1. Clipping rule $\alpha_t$: in most settings, $\alpha_t = \min\left\{1, \frac{c}{\|g_t^c\|}\right\}$ for some stochastic gradient $g_t^c$ and threshold $c$.

2. "Unclipped" step size $\eta_t$: the product of $\eta_t$ and $\alpha_t$ constitutes the SGD step size.

3. Threshold $c$: we say that clipping occurs whenever $\|g_t^c\| \geq c$. We intentionally define this separately from $\alpha_t$ to allow applications of the algorithm to set $\alpha_t = 1$.

A key aspect of Algorithm 1 is that it uses "double sampling," querying the oracle twice in each iteration. The two queries are on the same point but are independent from each other. One sample is used to compute the "unclipped" step size $\eta_t$ and the clipping rule $\alpha_t$, and the other sample determines the direction of the gradient step. This enables analyzing $\mathbb{E}[\eta_t \alpha_t g_t]$ by conditioning on the clipping result without incurring a bias in $g_t$. We remark that this method is also used in Yang et al. [41], who analyze normalized SGD for non-convex functions.

---

[4]The twice-differentiability assumption can be relaxed by using a smoothness definition such as in Koloskova et al. [15], for which the smoothness lemmas we rely on still apply.

---

**Algorithm 1:** Clipped SGD With Double Sampling

---

**Input:** Initialization $x_0 \in \mathbb{R}^d$, gradient oracle $\mathcal{G}$ and bound $R$ on $\|x_0 - x^\star\|$.

**Parameters:** Clipping rule $\alpha_t$, "unclipped" step size $\eta_t$ and threshold $c$.

**1** $\mathcal{T}_1, \mathcal{T}_2 \leftarrow \emptyset$

**2 for** $t = 0, 1, 2, \ldots$ **do**

**3**     $g_t^c \leftarrow \mathcal{G}(x_t)$

**4**     compute $\alpha_t$ and $\eta_t$ using $g_t^c$

**5**     **if** $c \leq \|g_t^c\|$ **then** $\mathcal{T}_1 = \mathcal{T}_1 \cup \{t\}$ **else** $\mathcal{T}_2 = \mathcal{T}_2 \cup \{t\}$

**6**     $g_t \leftarrow \mathcal{G}(x_t)$

**7**     $x_{t+1} = \mathrm{Proj}_{\mathbb{B}(x_0, R)}(x_t - \eta_t \alpha_t g_t)$     ▷ projection only required for variants of Adaptive SGD

**8 if** $|\mathcal{T}_2| \neq \emptyset$ **then** $\bar{x} = \frac{1}{|\mathcal{T}_2|} \sum_{t \in \mathcal{T}_2} x_t$ **else** $\bar{x} = x_0$

**9 return** $\bar{x}$

---

Another non-standard aspect of Algorithm 1 is the way it chooses which iterates to average for the final result. The algorithm tracks the sets $\mathcal{T}_1$ and $\mathcal{T}_2$, which correspond to iterations where clipping occurs/does not occur, respectively. These sets are considered when deciding the return value: if $\mathcal{T}_2 \neq \emptyset$ then we return $\bar{x} = \frac{1}{|\mathcal{T}_2|} \sum_{t \in \mathcal{T}_2} x_t$, and otherwise we return the initial point $\bar{x} = x_0$; our proofs show that $|\mathcal{T}_2| \geq \frac{T}{2}$ with high probability.

Table 1: Definition of our methods as applications of Algorithm 1. Under Assumption 3 (bounded noise) we set $\sigma' := \sigma$. Under Assumption 4 (light tails) we set $\sigma' := 3\sqrt{\log\left(\frac{T}{\delta}\right)}\sigma$.

| | **step size** $\eta_t$ | **clipping rule** $\alpha_t$ | **threshold** $c$ |
|---|---|---|---|
| standard | $\frac{1}{16} \min\left\{ \frac{1}{11L_0}, \frac{1}{L_0 + \frac{\sigma'\sqrt{T}}{R_0}} \right\}$ | $\min\left\{ 1, \frac{c}{\|g_t^c\|} \right\}$ | $\frac{1}{L_1} \max\left\{ 10L_0, \frac{\sqrt{T}}{R_0}\sigma' \right\}$ |
| implicit | $\frac{1}{8}\left( L_0 + \|g_t^c\|L_1 + \frac{\sigma'\sqrt{T}}{R_0} \right)^{-1}$ | $1$ | $\frac{1}{L_1} \max\left\{ 10L_0, \frac{\sqrt{T}}{R_0}\sigma' \right\}$ |
| conservative | $\frac{1}{16} \min\left\{ \frac{1}{11L_0}, \frac{1}{L_0 + \frac{\sigma'\sqrt{T}}{R_0}} \right\}$ | $\min\left\{ 1, \frac{c}{\|g_t^c\|} \right\}$ | $64\sqrt{\log_+\left(\frac{T}{\delta}\right)} \frac{R_0}{\sqrt{T}} \max\left\{ 10L_0, \frac{\sqrt{T}}{R_0}\sigma' \right\}$ |
| adaptive | $R\left( \sum_{i=0}^{t} \alpha_i^2 \|g_i\|^2 \right)^{-\frac{1}{2}}$ | $\min\left\{ 1, \frac{c}{\|g_t^c\|} \right\}$ | $\frac{1}{L_1} \max\left\{ 10L_0, \frac{\sqrt{T}}{R}\sigma' \right\}$ |
| adaptive + conservative | $R\left( \sum_{i=0}^{t} \alpha_i^2 \|g_i\|^2 \right)^{-\frac{1}{2}}$ | $\min\left\{ 1, \frac{c}{\|g_t^c\|} \right\}$ | $15\sqrt{\log_+\left(\frac{T}{\delta}\right)} \frac{R}{\sqrt{T}} \max\left\{ 10L_0, \frac{\sqrt{T}}{R}\sigma' \right\}$ |

Table 1 presents our different methods, that is, the different applications of Algorithm 1. We provide a short description of each method:

1. "Standard clipping" is an extension of the common clipping stepsizes from the deterministic $(L_0, L_1)$-smooth setting [44, 43, 38]. Indeed, for $\sigma = 0$, when ignoring constants, we have $\eta_t \alpha_t = \min\left\{ \frac{1}{L_0}, \frac{1}{L_1\|g_t^c\|} \right\}$.

2. "Implicit clipping" is the method that prior work refers to as a "normalized step size" or "smoothed clipping" [44, 11]. In this method, $\alpha_t = 1$ and $\eta_t$ is a function of $g_t^c$.

3. "Conservative clipping" is a method that is independent of $L_1$. This method stems from the proof of Theorem 1, which requires $T \geq \log_+\left(\frac{T}{\delta}\right)\left(64L_1 R_0\right)^2$, thereby limiting $L_1$. We use this to modify standard clipping by replacing $L_1$ with its limit.

4. "Adaptive clipping" is a version of adaptive SGD with two changes: clipping is applied according to $\alpha_t$, and the gradient norms in the denominator are also clipped using $\alpha_1, ..., \alpha_t$.

5. "Adaptive + conservative clipping" is a similar method that is independent of $L_1$.

## 3.1 Clipped SGD

**Theorem 1.** *Let $f : \mathbb{R}^d \to \mathbb{R}$ and suppose Assumptions 1, 2 and 4 hold. Let $\delta \in (0, 1/2)$ and let $\bar{x}$ be the output of Algorithm 1 when run for $T \geq \log_+\left(\frac{T}{\delta}\right)(64L_1R_0)^2$ steps under one of the first 3 rows of Table 1. Then with probability at least $1 - 2\delta$, the optimality gap $f(\bar{x}) - f(x^\star)$ is*

$$O\left(\frac{\log_+\left(\frac{T}{\delta}\right)\left(L_0 R_0^2 + \sqrt{\log\left(\frac{T}{\delta}\right)}\sigma R_0 \sqrt{T}\right)}{T}\right).$$

We remark that Theorem 1 also holds when $R = \infty$. That is, the projection is not required for the first 3 methods from Table 1.

**Proof sketch.**  In the sketch, we first prove the desired rate under Assumption 3 and with probability at least $1 - \delta$. To obtain the desired rate under Assumption 4 with probability at least $1 - 2\delta$, we apply a reduction from Attia and Koren [1], which we elaborate on at the end of the sketch.

We begin by presenting the main outline of the proof, in which we state two claims that will be proven immediately following the outline. To maintain conciseness, we defer the full details to Appendices C and D. We start with the first claim.

**Claim 1.** *With probability at least $1 - \delta$,*

$$\sum_{t=0}^{T-1} \eta_t \alpha_t \Delta_t \leq 2\log_+\left(\tfrac{T}{\delta}\right)R_0^2.$$

Separating the clipped and unclipped iterations, we get

$$\sum_{t \in \mathcal{T}_2} \eta_t \alpha_t \Delta_t \leq 2\log_+\left(\tfrac{T}{\delta}\right)R_0^2 - \sum_{t \in \mathcal{T}_1} \eta_t \alpha_t \Delta_t.$$

We now observe that clipped iterations make large progress, as stated in the following claim.

**Claim 2.** *If $t \in \mathcal{T}_1$ then $\eta_t \alpha_t \Delta_t \geq 4\log_+\left(\tfrac{T}{\delta}\right)R_0^2/T$.*

Therefore, with probability at least $1 - \delta$,

$$\sum_{t \in \mathcal{T}_2} \eta_t \alpha_t \Delta_t \leq 2\log_+\left(\tfrac{T}{\delta}\right)R_0^2 - 4\log_+\left(\tfrac{T}{\delta}\right)R_0^2 \frac{|\mathcal{T}_1|}{T}.$$

This implies $|\mathcal{T}_1| \leq \frac{T}{2}$ and therefore $|\mathcal{T}_2| \geq \frac{T}{2}$. In particular, this shows that the output of the algorithm is $\bar{x} = \frac{1}{|\mathcal{T}_2|}\sum_{t \in \tau_2} x_t$.

In unclipped iterations we have $\|g_t^c\| \leq c$. The norm $\|g_t^c\|$ appears only in the *denominator* of $\eta_t \alpha_t$, so we can bound $\eta_t \alpha_t$ from below by substituting $\|g_t^c\|$ with $c$. Thus, we show that $\eta_t \alpha_t \geq \frac{1}{16}\left(11L_0 + \frac{\sigma\sqrt{T}}{R_0}\right)^{-1} := \gamma$ (see details in Lemma 7). By this we have

$$\gamma \sum_{t \in \mathcal{T}_2} \Delta_t \leq \sum_{t \in \mathcal{T}_2} \eta_t \alpha_t \Delta_t \leq 2\log_+\left(\tfrac{T}{\delta}\right)R_0^2.$$

Dividing by $\gamma|\mathcal{T}_2|$ and using the bound on $|\mathcal{T}_2|$, we have

$$\frac{1}{|\mathcal{T}_2|}\sum_{t \in \mathcal{T}_2} \Delta_t \leq \frac{64\left(11L_0 + \frac{\sigma\sqrt{T}}{R_0}\right)\log_+\left(\tfrac{T}{\delta}\right)R_0^2}{T}.$$

Using Jensen's inequality completes the proof.

**Proof sketch of Claim 2** (full proof is in Lemmas 6 and 7). Consider iterations where $t \in \mathcal{T}_1$, that is, iterations where $\|g_t^c\| > c$: By our choice of threshold we have $c \geq 6\sigma$. Therefore, for the sample to be above the threshold, the gradient norm must dominate over the noise, implying $\|g_t^c\| \approx \|\nabla f(x_t)\|$. A known property of $(L_0, L_1)$-smooth functions is that for any $x \in \mathbb{R}^d$,

$$\|\nabla f(x)\|^2 \leq 2(L_0 + \|\nabla f(x)\|L_1)(f(x) - f(x^\star)).$$

Substituting $x = x_t$, using our choice of $\alpha_t$ and $\eta_t$, and substituting $\|g_t^c\| \approx \|\nabla f(x_t)\|$, we get

$$\|g_t^c\|^2 \leq 2(L_0 + \|g_t^c\|L_1)\Delta_t \leq (2\eta_t\alpha_t)^{-1}\Delta_t.$$

Multiplying by $2(\eta_t\alpha_t)^2$, we get

$$\eta_t\alpha_t\Delta_t \geq 2(\eta_t\alpha_t\|g_t^c\|)^2.$$

Clipping implies

$$\eta_t\alpha_t\|g_t^c\| \overset{(i)}{\approx} \eta_t c \overset{(ii)}{\geq} 2\sqrt{\log_+\left(\tfrac{T}{\delta}\right)}\frac{R_0}{\sqrt{T}},$$

where $(i)$ is an equality in the case $\alpha_t = \min\left\{1, \frac{c}{\|g_t^c\|}\right\}$ and $(ii)$ is due to the choice of $\eta_t$ and $c$.

Combining the last two inequalities, we get the bound $\eta_t\alpha_t\Delta_t \geq 8\log_+\left(\tfrac{T}{\delta}\right)\frac{R_0^2}{T}$.

**Proof sketch of Claim 1** (full proof is in Lemma 8). We split the signal from the noise by expressing the sum $\sum_{t=0}^{T-1} \eta_t\alpha_t\langle\nabla f(x_t), x_t - x^\star\rangle$ as

$$\underbrace{\sum_{t=0}^{T-1} \eta_t\alpha_t\langle g_t, x_t - x^\star\rangle}_{S_1} + \underbrace{\sum_{t=0}^{T-1} \eta_t\alpha_t\langle\nabla f(x_t) - g_t, x_t - x^\star\rangle}_{S_2}.$$

To bound $S_2$, we use techniques from Attia and Koren [1]. The random variables $\eta_t\alpha_t$ and $g_t$ are independent conditionally on $x_t$ due to the double sampling, and therefore the elements of $S_2$ form a *martingale difference sequence* w.r.t. $\xi_t = (x_t, g_t^c)$. This allows us to bound $S_2$ using a martingale concentration bound and standard analysis. We get that with probability at least $1 - \delta$,

$$S_2 \leq \left(\frac{1}{8} + \frac{5}{16}\log\left(\frac{T}{\delta}\right)\right)R_0^2 + \frac{1}{8}\sum_{t=0}^{T-1}\eta_t^2\alpha_t^2\|g_t\|^2.$$

To bound $S_1$, we use standard analysis and show that

$$S_1 \leq \frac{R_0^2}{2} + \frac{1}{2}\sum_{t=0}^{T-1}\eta_t^2\alpha_t^2\|g_t\|^2.$$

By the convexity of $f$ and the above displays we find that, with probability at least $1 - \delta$,

$$\sum_{t=0}^{T-1}\eta_t\alpha_t\Delta_t \leq \sum_{t=0}^{T-1}\eta_t\alpha_t\langle\nabla f(x_t), x_t - x^\star\rangle$$

$$\leq S_1 + S_2 \leq \frac{1}{2}\log_+\left(\tfrac{T}{\delta}\right)R_0^2 + \frac{3}{4}\sum_{t=0}^{T-1}\eta_t^2\alpha_t^2\|g_t\|^2.$$

To bound $\eta_t^2\alpha_t^2\|g_t\|^2$, we consider the cases of high noise and low noise: when $\|g_t\| \geq 6\sigma$, like in clipped iterations, we have $\|g_t\| \approx \|\nabla f(x_t)\|$ and therefore $\eta_t^2\alpha_t^2\|g_t\|^2 \leq \frac{1}{2}\eta_t\alpha_t\Delta_t$. when $\|g_t\| \leq 6\sigma$, we use $\eta_t\alpha_t \leq \frac{R_0}{8\sigma\sqrt{T}}$ and get $\eta_t^2\alpha_t^2\|g_t\|^2 \leq \frac{R_0^2}{T}$.

Plugging everything in and rearranging, we have that with probability at least $1 - \delta$,

$$\sum_{t=0}^{T-1}\eta_t\alpha_t\Delta_t \leq 2\log_+\left(\tfrac{T}{\delta}\right)R_0^2.$$

**Obtaining the result for light-tailed noise.** Let $\mathcal{G}$ be an unbiased gradient oracle with $\sigma$-sub-Gaussian noise. Attia and Koren [1, Appendix A] show that there exists an unbiased gradient oracle $\tilde{\mathcal{G}}$ with $3\sqrt{\log\left(\frac{T}{\delta}\right)}\sigma$-bounded noise that, with probability at least $1 - \delta$, has the exact same output as $\mathcal{G}$ throughout the entire algorithm. Therefore, with probability at least $1 - \delta$, we have the same guarantee as when assuming $3\sqrt{\log\left(\frac{T}{\delta}\right)}\sigma$-bounded noise. By using a union bound, we get that the desired guarantee holds under $\sigma$-sub-Gaussian noise with probability at least $1 - 2\delta$.

## 3.2 Clipped Adaptive SGD

**Theorem 2.** *Assume the setting of Theorem 1 under one of the last 2 rows of Table 1. Then with probability at least $1 - 2\delta$, the optimality gap $f(\bar{x}) - f(x^\star)$ is*

$$O\left(\frac{\log_+\left(\frac{1}{\delta}\right)\left(L_0 R^2 + \sqrt{\log\left(\frac{T}{\delta}\right)}\sigma R\sqrt{T}\right)}{T}\right).$$

The proof shares the main ideas of the proof of Theorem 1. The main difference is that analyses of AdaGrad-like stepsizes handle the stepsize in a very specific manner. In our case, use it we show that

$$\sum_{t=0}^{T-1} \alpha_t \langle g_t, x_t - x^\star \rangle \leq 2R\sqrt{\sum_{i=0}^{T-1} \alpha_i^2 \|g_i\|^2}.$$

Therefore, for the rest of the proof, we analyze $\sum_{t=0}^{T-1} \alpha_t \Delta_t$ instead of $\sum_{t=0}^{T-1} \eta_t \alpha_t \Delta_t$.

## 4 Experiments

Our work introduces several non-standard algorithmic choices that facilitate our theoretical analysis. We conduct experiments in order to assess the empirical effect of those choices. Specifically, we aim to shed some light on the following questions:

1. Does gradient clipping help in stochastic, convex, $(L_0, L_1)$-smooth optimization?
2. Is double-sampling better than single sampling?
3. Does the average of iterates from $\mathcal{T}_2$ perform better than the average of all iterates?
4. How does "adaptive clipping" compare to "standard clipping"? How does it compare to adaptive SGD with no clipping?

We perform linear regression on the California Housing dataset [28] and the Parkinsons Telemonitoring dataset [36] (the latter is in Appendix H) using the loss function $f(w) = \|Xw - y\|^4$. For algorithms with a fixed stepsize, we set $\eta$ to a variable $lr$ which we tune. For algorithms with a time-dependent stepsize, we express $\eta_t$ as a function of the clipping threshold $c$ and multiply the result by a factor of $lr$ which we tune. For each tested method, we tune both $lr$ and $c$ (when applicable) using a two-level, two-dimensional grid search. We defer to Appendix H for additional details on the definitions of $\eta_t$ and the tuning process. We also perform similar synthetic experiments on a function of the form $f(w) = \|Aw\|^4$ (Appendix H). The code for reproducing the experiments is available at github.com/formll/clipped-sgd-under-generalized-smoothness.

**Comparison of clipping methods.** Figure 1a compares the output of Algorithm 1 for the methods of standard, implicit and adaptive clipping (rows 1, 2 and 4 in Table 1). The three methods show similar dynamics and converge to a nearly identical optimality gap.

Figures 1b and 1c compare SGD, adaptive SGD, Algorithm 1 with standard clipping (clipped SGD) and Algorithm 1 with adaptive clipping (clipped adaptive SGD). For SGD and adaptive SGD, we plot the sub-optimality of the average of *all* iterates, set $\alpha_t = 1$ and set $\eta_t$ as in their clipped counterparts. The figure shows overall similar performance. In SGD the clipped method performs a bit worse. In Adaptive SGD the difference between clipping and no clipping is more substantial, in favor of clipping.

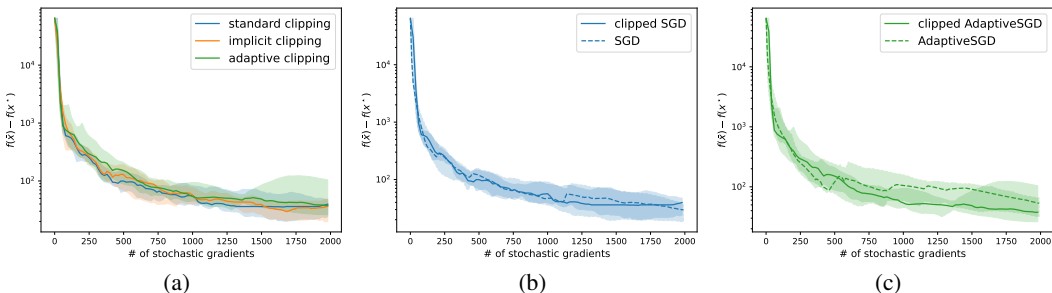

Figure 1: Sub-optimality of SGD variants as a function of the number of stochastic gradients used, when training a quartic-loss linear regression model on the California Housing dataset. We plot the median across 10 runs, with a shaded region showing the inter-quartile range.

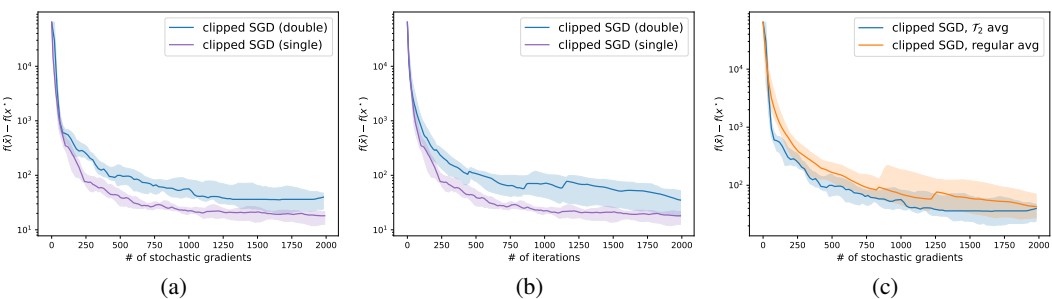

Figure 2: Ablations of Algorithm 1. Figures 2a and 2b compares single and double sampling by plotting sub-optimality as a function of gradient and iteration budget, respectively. Figure 2c compares different averaging methods. We plot the median across 10 runs and shade the inter-quartile range.

**Comparison of theory with empirical results.**    We test the effect of our double-sampling approach, as it originated from analytical considerations and not from practice. Figure 2a plots the sub-optimality of standard clipping in two versions: one as presented in the paper, and another that uses a *single* sample in each iteration. Note that the x-axis is the number of stochastic gradients, so the latter version ran for twice as many iterations. There seem to be no advantage to double-sampling, suggesting it might not be necessary in order to prove convergence in the stochastic convex regime. Figure 2b plots a comparison in terms of iteration complexity, where single sampling still achieves better sub-optimality.

We move on to investigate our algorithmic choice of defining the output as $\bar{x} = \frac{1}{|\mathcal{T}_2|} \sum_{t \in \mathcal{T}_2} x_i$. Figure 2c compares the sub-optimality of $\bar{x}$ to the sub-optimality of the average across all iterates (both are with standard clipping). We see that $\bar{x}$ achieves slightly better sub-optimality, supporting our choice.

## 5   Conclusion

In this paper, we analyze stochastic gradient descent with gradient clipping on convex, $(L_0, L_1)$-smooth functions. We prove a high-probability convergence rate for clipped SGD, and introduce a clipped variation of adaptive SGD that has a similar rate.

There are various possible directions for future work. First, since the double-sampling approach is not supported by empirical data, it is interesting to study convergence without it. Another direction is extending our analysis to a more generalized smoothness assumption such as $\ell$-smoothness [18]. Lastly, exploring tuning-free methods that require no knowledge on problem parameters could be of both theoretical and practical interest.

## Acknowledgments

This research was partially supported by the Israeli Science Foundation (ISF) grant no. 2486/21 and the Adelis Foundation. It was also partially supported by Israel PBC-VATAT, by the Technion Artificial Intelligence Hub (Tech.AI), and by the Israel Science Foundation (grant No. 3109/24).

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

# A Lemmas for $(L_0, L_1)$-Smooth Functions

**Lemma 1.** *Let $f : \mathbb{R}^d \to \mathbb{R}$ and suppose Assumption 2 holds. Then for any $x \in \mathbb{R}^d$,*

$$\|\nabla f(x)\|^2 \leq 2(L_0 + L_1 \|\nabla f(x)\|)(f(x) - f(x^\star)).$$

Koloskova et al. [15, Lemma A.2] prove this by simply using Zhang et al. [43, Lemma A.3], which assumes the $(L_0, L_1)$-smoothness definition we use.

**Lemma 2.** *Let $f : \mathbb{R}^d \to \mathbb{R}$ and suppose Assumption 2 holds. Then for any $x \in \mathbb{R}^d$,*

$$\|\nabla f(x)\| \leq \max\left\{ 3L_1(f(x) - f(x^\star)), 6\frac{L_0}{L_1} \right\}.$$

*Proof.* Denote $\Delta := f(x) - f(x^\star)$. Lemma 1 shows that

$$\|\nabla f(x)\|^2 - 2L_1\Delta\|\nabla f(x)\| - 2L_0\Delta \leq 0.$$

This is a quadratic inequality in $\|\nabla f(x)\|$. Since $L_0, L_1, \Delta$ and $\|\nabla f(x)\|$ are non-negative, the solution is less than the parabola's largest root. Therefore,

$$\|\nabla f(x)\| \leq \frac{1}{2}\left( 2L_1\Delta + \sqrt{4L_1^2\Delta^2 + 8L_0\Delta} \right) = L_1\Delta + \sqrt{L_1^2\Delta^2 + 2L_0\Delta} \leq 2L_1\Delta + \sqrt{2L_0\Delta}.$$

If $L_1^2\Delta^2 \geq 2L_0\Delta$ then we get

$$\|\nabla f(x)\| \leq 2L_1\Delta + \sqrt{L_1^2\Delta^2} = 3L_1\Delta.$$

If $L_1^2\Delta^2 < 2L_0\Delta$: Without loss of generality, we assume $\Delta > 0$ (since for $\Delta = 0$ the result is immediate from Lemma 1), and therefore $\Delta < 2L_0/L_1^2$. Consequently,

$$\|\nabla f(x)\| \leq 2\sqrt{2L_0\Delta} + \sqrt{2L_0\Delta} = 3\sqrt{2L_0\Delta} \leq 3\sqrt{4L_0^2/L_1^2} = 6L_0/L_1.$$

Overall, we have

$$\|\nabla f(x_t)\| \leq \max\left\{ 3L_1\Delta, 6\frac{L_0}{L_1} \right\}$$

$\square$

# B Lemmas on Probability

To achieve high probability bounds, we use the following concentration inequality, which is a corollary of Li and Orabona [19, Lemma 1].

**Lemma 3.** *Assume that $Z_1, Z_2, ..., Z_T$ is a martingale difference sequence with respect to $\xi_1, \xi_2, ..., \xi_T$ (i.e., $\mathbb{E}[Z_t \mid \xi_1, \ldots, \xi_{t-1}] = 0$) and that $|Z_t| \leq \sigma_t$ for all $1 \leq t \leq T$, where $\sigma_t$ is a sequence of random variables such that $\sigma_t$ is measurable with respect to $\xi_1, \xi_2, \ldots, \xi_{t-1}$. Then, for any fixed $\lambda > 0$ and $\delta \in (0, 1)$, with probability at least $1 - \delta$, we have*

$$\sum_{t=1}^{T} Z_t \leq \frac{3}{4}\lambda \sum_{t=1}^{T} \sigma_t^2 + \frac{1}{\lambda} \ln \frac{1}{\delta}.$$

To handle the light tails assumption, we use a reduction from sub-Gaussian noise to bounded noise presented by Attia and Koren [1, Appendix A]. The reduction can be formally stated as follows.

**Lemma 4.** *Let $\mathcal{G}$ be an unbiased oracle satisfying Assumption 4. Then for any $x_0, ..., x_{T-1}$ and any $\delta \in (0, 1)$, there exists an unbiased oracle $\tilde{\mathcal{G}}$ such that*

*(i) $\tilde{\mathcal{G}}$ satisfies Assumption 3 for $\tilde{\sigma} := 3\sigma\sqrt{\log\left(\frac{T}{\delta}\right)}$.*

*(ii) With probability at least $1 - \delta$, for all $t = 0, \ldots, T - 1$ it holds that $\tilde{\mathcal{G}}(x_t) = \mathcal{G}(x_t)$.*

## C Lemmas for proving Theorem 1

**Lemma 5.** *The value of $\eta_t \alpha_t$ is always smaller under "standard clipping" than under "implicit clipping." Additionally, if $2\log_+\left(\frac{T}{\delta}\right)(64L_1R_0)^2 \leq T$, then it is always smaller under "conservative clipping" than under "standard clipping."*

*Proof.* First, let us compare conservative clipping and standard clipping: The definition of $\eta_t$ is equivalent in both methods. Due to the assumption on $T$, the threshold $c$ is smaller in conservative clipping than in standard clipping. This immediately leads to the stated relationship regarding $\eta_t \alpha_t$.

Now let us compare standard clipping and implicit clipping: For the rest of the proof, let $\eta_t$, $\alpha_t$ and $c$ be defined as in standard clipping. Observe that

$$\eta_t = \min\left\{\frac{1}{11L_0}, \frac{1}{L_0 + \frac{\sigma\sqrt{T}}{R_0}}\right\} = \frac{1}{L_0 + \max\left\{10L_0, \frac{\sigma\sqrt{T}}{R_0}\right\}} = \frac{1}{L_0 + cL_1}$$

and

$$\eta_t = \min\left\{\frac{1}{11L_0}, \frac{1}{L_0 + \frac{\sigma\sqrt{T}}{R_0}}\right\} = \min\left\{\frac{1}{11L_0}, \frac{1}{L_0 + \frac{\sigma\sqrt{T}}{R_0}}, \frac{1}{\frac{\sigma\sqrt{T}}{R_0}}\right\} = \min\left\{\eta_t, \frac{R_0}{\sigma\sqrt{T}}\right\}.$$

Together we get

$$\eta_t = \min\left\{\frac{1}{L_0 + cL_1}, \frac{R_0}{\sigma\sqrt{T}}\right\}.$$

Consider the case of unclipped iterations, which satisfy $\alpha_t = 1$ and $c > \|g_t^c\|$. In this case,

$$\eta_t \alpha_t = \eta_t = \frac{1}{16}\min\left\{\frac{1}{L_0 + cL_1}, \frac{R_0}{\sigma\sqrt{T}}\right\} \leq \frac{1}{16}\min\left\{\frac{1}{L_0 + \|g_t^c\|L_1}, \frac{R_0}{\sigma\sqrt{T}}\right\}.$$

Now consider the case of clipped iterations, which satisfy $\alpha_t = \frac{c}{\|g_t^c\|}$ and $c \leq \|g_t^c\|$. In this case,

$$\begin{aligned}
\eta_t \alpha_t &= \frac{1}{16}\alpha_t \min\left\{\frac{1}{L_0 + cL_1}, \frac{R_0}{\sigma\sqrt{T}}\right\} \\
&\leq \frac{1}{16}\alpha_t \min\left\{\frac{1}{(c/\|g_t^c\|)L_0 + cL_1}, \frac{R_0}{\sigma\sqrt{T}}\right\} \\
&= \frac{1}{16}\alpha_t \min\left\{\frac{1}{(c/\|g_t^c\|)L_0 + \|g_t^c\|L_1}, \frac{R_0}{\sigma\sqrt{T}}\right\} \\
&\leq \frac{1}{16}\min\left\{\alpha_t\frac{1}{(c/\|g_t^c\|)L_0 + \|g_t^c\|L_1}, \frac{R_0}{\sigma\sqrt{T}}\right\} \\
&= \frac{1}{16}\min\left\{\frac{1}{L_0 + \|g_t^c\|L_1}, \frac{R_0}{\sigma\sqrt{T}}\right\}.
\end{aligned}$$

Therefore, both cases satisfy

$$\begin{aligned}
\eta_t \alpha_t = \frac{1}{16}\min\left\{\frac{1}{L_0 + \|g_t^c\|L_1}, \frac{1}{\frac{\sigma\sqrt{T}}{R_0}}\right\} &= \frac{1}{16}\frac{1}{\max\left\{L_0 + \|g_t^c\|L_1, \frac{\sigma\sqrt{T}}{R_0}\right\}} \\
&\leq \frac{1}{16}\frac{1}{\frac{1}{2}\left(L_0 + \|g_t^c\|L_1 + \frac{\sigma\sqrt{T}}{R_0}\right)} = \frac{1}{8}\frac{1}{L_0 + \|g_t^c\|L_1 + \frac{\sigma\sqrt{T}}{R_0}}.
\end{aligned}$$

$\square$

**Lemma 6.** *Let $f : \mathbb{R}^d \to \mathbb{R}$ and suppose Assumptions 1 to 3 hold. Additionally, assume that $\eta_t \alpha_t \leq \frac{1}{8}\left(L_0 + \|g_t^c\|L_1 + \frac{\sigma\sqrt{T}}{R_0}\right)^{-1}$ and $6\sigma \leq c \leq \|g_t^c\|$. Then for any $g \in \mathcal{G}(x_t)$,*

$$\eta_t^2 \alpha_t^2 \|g\|^2 \leq \frac{1}{2}\eta_t \alpha_t \Delta_t.$$

*Proof.* By our assumptions, we have $6\sigma \leq \|g_t^c\|$. This implies $\sigma \leq (1/5)\|\nabla f(x_t)\|$, as otherwise

$$\|g_t^c\| \leq \|g_t^c - \nabla f(x_t)\| + \|\nabla f(x_t)\| < \sigma + 5\sigma = 6\sigma.$$

Therefore, every oracle query satisfies

$$\|\nabla f(x_t) - \mathcal{G}(x_t)\| \leq \sigma \leq (1/5)\|\nabla f(x_t)\|.$$

By the triangle inequalities, we obtain

$$(4/5)\|\nabla f(x_t)\| \leq \|\mathcal{G}(x_t)\| \leq (6/5)\|\nabla f(x_t)\|. \tag{1}$$

Therefore,

$$(\eta_t \alpha_t)^{-1} \geq 8\Big(L_0 + \|g_t^c\|L_1 + \tfrac{\sigma\sqrt{T}}{R}\Big) \geq 8(L_0 + \|g_t^c\|L_1) \tag{2}$$

$$\overset{(1)}{\geq} 8(L_0 + (4/5)\|\nabla f(x_t)\|L_1) \geq 8(4/5)(L_0 + \|\nabla f(x_t)\|L_1).$$

By applying the smoothness property stated in Lemma 1, we obtain

$$\eta_t^2 \alpha_t^2 \|\mathcal{G}_t(x_t)\|^2 \overset{(2)}{\leq} \frac{1}{8(4/5)}\eta_t \alpha_t \left(\frac{1}{L_0 + \|\nabla f(x_t)\|L_1}\right)\|\mathcal{G}_t(x_t)\|^2$$

$$\overset{(1)}{\leq} \frac{(6/5)^2}{8(4/5)}\eta_t \alpha_t \left(\frac{1}{L_0 + \|\nabla f(x_t)\|L_1}\right)\|\nabla f(x_t)\|^2 \leq \frac{1}{2}\eta_t \alpha_t \Delta_t.$$

$\square$

**Lemma 7.** *Assume that the expression $\eta_t \alpha_t$ is between its value under "conservative clipping" and its value under "implicit clipping". Additionally, assume that the threshold $c$ is no less than its value under "conservative clipping". Then*

*(i) $c \geq \|g_t^c\|$ implies $\eta_t \alpha_t \geq \frac{1}{16}\Big(11L_0 + \tfrac{\sigma\sqrt{T}}{R_0}\Big)^{-1}$;*

*(ii) $c \leq \|g_t^c\|$ implies $\eta_t \alpha_t \Delta_t \geq 8\log_+\big(\tfrac{T}{\delta}\big)\tfrac{R_0^2}{T}$.*

*Proof.* Consider iterations where $c \geq \|g_t^c\|$. In this case, under the choice of $\eta_t$ and $\alpha_t$ in "conservative clipping", we have

$$\eta_t \alpha_t = \eta_t = \frac{1}{16}\min\left\{\frac{1}{11L_0}, \frac{1}{L_0 + \tfrac{\sigma\sqrt{T}}{R_0}}\right\} \geq \frac{1}{16}\frac{1}{11L_0 + \tfrac{\sigma\sqrt{T}}{R_0}}.$$

Due to Lemma 5, this lower bound applies to any choice of $\eta_t$ and $\alpha_t$ that matches our setting.

Consider iterations where $c \leq \|g_t^c\|$. In this case we have $\sigma \leq (1/5)\|\nabla f(x_t)\|$: otherwise,

$$\|g_t^c\| \leq \|g_t^c - \nabla f(x_t)\| + \|\nabla f(x_t)\| < \sigma + 5\sigma = 6\sigma \leq c \leq \|g_t^c\|,$$

where the last two inequalities are by our assumptions on $c$. Therefore Lemma 6 applies, and therefore

$$\frac{1}{2}\eta_t \alpha_t \Delta_t \overset{(i)}{\geq} \eta_t^2 \alpha_t^2 \|g_t^c\|^2$$

$$\overset{(ii)}{\geq} \left(\frac{1}{16}\min\left\{\frac{1}{11L_0}, \frac{1}{L_0 + \tfrac{\sigma\sqrt{T}}{R_0}}\right\}\min\left\{1, \frac{c}{\|g_t^c\|}\right\}\|g_t^c\|\right)^2$$

$$\overset{(iii)}{=} \left(\frac{1}{16}\min\left\{\frac{1}{11L_0}, \frac{1}{L_0 + \tfrac{\sigma\sqrt{T}}{R_0}}\right\}\cdot c\right)^2$$

$$\overset{(iv)}{\geq} \left(\frac{1}{16}\min\left\{\frac{1}{11L_0}, \frac{1}{L_0 + \tfrac{\sigma\sqrt{T}}{R_0}}\right\}\cdot 64\sqrt{\log_+\big(\tfrac{T}{\delta}\big)}\frac{R_0}{\sqrt{T}}\max\left\{10L_0, \frac{\sigma\sqrt{T}}{R_0}\right\}\right)^2,$$

where $(i)$ uses Lemma 6, $(ii)$ uses the assumption on $\eta_t \alpha_t$, $(iii)$ uses $c \leq \|g_t^c\|$ and $(iv)$ uses the value of $c$ under "conservative clipping".

From this point we are done with our assumptions and only use pure algebra:

$$\cdots \geq \left( \frac{1}{16} \min\left\{ \frac{1}{11L_0}, \frac{1}{L_0 + \frac{\sigma\sqrt{T}}{R_0}} \right\} \cdot 64 \sqrt{\log_+\left(\frac{T}{\delta}\right)} \frac{R_0}{\sqrt{T}} \max\left\{ 10L_0, \frac{\sigma\sqrt{T}}{R_0} \right\} \right)^2$$

$$= 16 \log_+\left(\tfrac{T}{\delta}\right) \frac{R_0^2}{T} \left( \min\left\{ \frac{1}{11L_0}, \frac{1}{L_0 + \frac{\sigma\sqrt{T}}{R_0}} \right\} \max\left\{ 10L_0, \frac{\sigma\sqrt{T}}{R_0} \right\} \right)^2$$

$$\geq 16 \log_+\left(\tfrac{T}{\delta}\right) \frac{R_0^2}{T} \left( \min\left\{ \frac{10L_0}{11L_0}, \frac{\frac{1}{2}\left(10L_0 + \frac{\sigma\sqrt{T}}{R_0}\right)}{L_0 + \frac{\sigma\sqrt{T}}{R_0}} \right\} \right)^2$$

$$\geq 16 \log_+\left(\tfrac{T}{\delta}\right) \frac{R_0^2}{T} \left( \min\left\{ \frac{10}{11}, \frac{1}{2} \right\} \right)^2$$

$$= 4 \log_+\left(\tfrac{T}{\delta}\right) \frac{R_0^2}{T}.$$

$\square$

**Lemma 8.** *Let $f : \mathbb{R}^d \to \mathbb{R}$ and suppose Assumptions 1 to 3 hold. Assume Algorithm 1 is run with parameters satisfying $\eta_t \alpha_t \leq \frac{1}{8}\left( L_0 + \|g_t^c\|L_1 + \frac{\sigma\sqrt{T}}{R_0} \right)^{-1}$. Then for any $\delta \in (0,1)$, with probability at least $1 - \delta$,*

$$\sum_{t=0}^{T-1} \eta_t \alpha_t \Delta_t \leq 2 \log_+\left(\tfrac{T}{\delta}\right) R_0^2.$$

*Proof.* We remark that the first section of the proof follows techniques used in Attia and Koren [1, Lemma 2].

Define $Z_t := \eta_t \alpha_t \langle \nabla f(x_t) - g_t, x_t - x^\star \rangle$. Recall from Algorithm 1 that $\eta_t$ and $\alpha_t$ are deterministic given $g_t^c$. Then the sequence $Z_0, \ldots, Z_{T-1}$ is a *martingale difference sequence* with respect to $\xi_{i-1} := (x_i, g_i^c)$:

$$\mathbb{E}[Z_t \mid x_t, g_t^c] = \eta_t \alpha_t \langle \nabla f(x_t) - \mathbb{E}[g_t \mid x_t, g_t^c], x_t - x^\star \rangle = 0.$$

Define $R_{\max,t} := \max_{0 \leq s \leq t} \{R_s\}$. Each $Z_t$ satisfies

$$|Z_t| \leq \eta_t \alpha_t \|\nabla f(x_t) - g_t\|\|x_t - x^\star\| \leq \eta_t \alpha_t \sigma R_t \leq \frac{1}{8}\left( \frac{\sigma\sqrt{T}}{R_0} \right)^{-1} \sigma R_{\max,t} \leq \frac{R_0 R_{\max,t}}{8\sqrt{T}}.$$

Therefore, by Lemma 3, for any $t \in [T]$, $\lambda > 0$ and $\delta \in (0,1)$, with probability at least $1 - \frac{\delta}{T}$ it holds that

$$\sum_{s=0}^{t-1} \eta_s \alpha_s \langle \nabla f(x_s) - g_s, x_s - x^\star \rangle \leq \frac{3}{4}\lambda \sum_{s=0}^{t-1} \frac{R_0^2 R_{\max,s}^2}{64T} + \frac{1}{\lambda} \log\left( \frac{T}{\delta} \right)$$

$$\leq \frac{1}{64}\lambda R_0^2 R_{\max,t-1}^2 + \frac{1}{\lambda} \log\left( \frac{T}{\delta} \right).$$

By applying a union bound and choosing $\lambda := 4R_0^{-2}$, we get that with probability at least $1 - \delta$,

$$\forall t = 1, \ldots, T : \sum_{s=0}^{t-1} \eta_s \alpha_s \langle \nabla f(x_s) - g_s, x_s - x^\star \rangle \leq \frac{1}{16} R_{\max,t-1}^2 + \frac{1}{4} \log\left( \frac{T}{\delta} \right) R_0^2. \quad (3)$$

Define $C := 2\left( \left(1 + \frac{1}{2}\log\left(\frac{T}{\delta}\right)\right) R_0^2 + \sum_{t=0}^{T-1} \eta_t^2 \alpha_t^2 \|g_t\|^2 \right)$. We show by induction on $t$ that, when (3) holds, $R_{\max,t}^2 \leq C$ for all $0 \leq t < T$. For $t = 0$ we have $R_{\max,0}^2 = R_0^2 \leq C$. Let us assume correctness for $0, \ldots, t-1$. This implies $R_{\max,t-1}^2 \leq C$, so to prove the induction step, it suffices to

show $R_t^2 \leq C$. By unfolding the definitions of $R_0, \ldots, R_t$ and using common projection algebra, we have

$$R_t^2 \leq R_0^2 - 2\sum_{s=0}^{t-1} \eta_s \alpha_s \langle g_s, x_s - x^\star \rangle + \sum_{s=0}^{t-1} \eta_s^2 \alpha_s^2 \|g_s\|^2$$

$$\overset{(i)}{\leq} R_0^2 - 2\sum_{s=0}^{t-1} \eta_s \alpha_s \langle g_s, x_s - x^\star \rangle + \sum_{s=0}^{t-1} \eta_s^2 \alpha_s^2 \|g_s\|^2 + 2\sum_{s=0}^{t-1} \eta_s \alpha_s \langle \nabla f(x_s), x_s - x^\star \rangle$$

$$= R_0^2 + 2\sum_{s=0}^{t-1} \eta_s \alpha_s \langle \nabla f(x_s) - g_s, x_s - x^\star \rangle + \sum_{s=0}^{t-1} \eta_s^2 \alpha_s^2 \|g_s\|^2$$

$$\overset{(3)}{\leq} \frac{1}{8} R_{\max,t-1}^2 + \left(1 + \frac{1}{2}\log\left(\frac{T}{\delta}\right)\right) R_0^2 + \sum_{s=0}^{t-1} \eta_s^2 \alpha_s^2 \|g_s\|^2$$

$$\leq \frac{1}{8} R_{\max,t-1}^2 + \frac{C}{2}$$

$$\leq C,$$

where $(i)$ holds since, by convexity and the optimality of $x^\star$,

$$0 \leq -(f(x^\star) - f(x_s)) \leq -\langle \nabla f(x_s), x^\star - x_s \rangle = \langle \nabla f(x_s), x_s - x^\star \rangle.$$

By applying the bound on $R_{\max,t}^2$ to Equation (3) we obtain that

$$\sum_{t=0}^{T-1} \eta_t \alpha_t \langle \nabla f(x_t) - g_t, x_t - x^\star \rangle \leq \frac{1}{8}\left(\left(1 + \frac{1}{2}\log\left(\frac{T}{\delta}\right)\right) R_0^2 + \sum_{t=0}^{T-1} \eta_t^2 \alpha_t^2 \|g_t\|^2\right) + \frac{1}{4}\log\left(\frac{T}{\delta}\right) R_0^2$$

$$= \left(\frac{1}{8} + \frac{5}{16}\log\left(\frac{T}{\delta}\right)\right) R_0^2 + \frac{1}{8}\sum_{t=0}^{T-1} \eta_t^2 \alpha_t^2 \|g_t\|^2.$$

Moving on, a standard analysis shows that

$$\sum_{t=0}^{T-1} \eta_t \alpha_t \langle g_t, x_t - x^\star \rangle \leq \frac{R_0^2}{2} + \frac{1}{2}\sum_{t=0}^{T-1} \eta_t^2 \alpha_t^2 \|g_t\|^2.$$

By summing the two inequalities and using convexity, we get

$$\sum_{t=0}^{T-1} \eta_t \alpha_t \Delta_t \leq \sum_{t=0}^{T-1} \eta_t \alpha_t \langle \nabla f(x_t), x_t - x^\star \rangle$$

$$\leq \left(\frac{5}{8} + \frac{5}{16}\log\left(\frac{T}{\delta}\right)\right) R_0^2 + \frac{3}{4}\sum_{t=0}^{T-1} \eta_t^2 \alpha_t^2 \|g_t\|^2$$

$$\leq \frac{1}{2}\log_+\left(\frac{T}{\delta}\right) R_0^2 + \frac{3}{4}\sum_{t=0}^{T-1} \eta_t^2 \alpha_t^2 \|g_t\|^2.$$

When $\sigma \geq (1/5)\|\nabla f(x_t)\|$, we have

$$\|g_t\| \leq \|g_t - \nabla f(x_t)\| + \|\nabla f(x_t)\| \leq \sigma + 5\sigma = 6\sigma$$

and therefore

$$\eta_t^2 \alpha_t^2 \|g_t\|^2 \leq (6\sigma \eta_t \alpha_t)^2 \leq \left(\frac{6\sigma}{8}\left(\frac{\sigma\sqrt{T}}{R_0}\right)^{-1}\right)^2 \leq \frac{R_0^2}{T}.$$

When $\sigma < (1/5)\|\nabla f(x_t)\|$, Lemma 6 shows that

$$\eta_t^2 \alpha_t^2 \|g_t\|^2 \leq \frac{1}{2}\eta_t \alpha_t \Delta_t.$$

Using these two inequalities, we get

$$\sum_{t=0}^{T-1} \eta_t \alpha_t \Delta_t \leq \frac{1}{2} \log_+\left(\frac{T}{\delta}\right) R_0^2 + \frac{3}{4} \sum_{t=0}^{T-1} \left(\frac{1}{2}\eta_t \alpha_t \Delta_t + \frac{R_0^2}{T}\right)$$

$$\leq \frac{1}{2} \log_+\left(\frac{T}{\delta}\right) R_0^2 + \frac{1}{2} \sum_{t=0}^{T-1} \eta_t \alpha_t \Delta_t + R_0^2$$

$$\leq \log_+\left(\frac{T}{\delta}\right) R_0^2 + \frac{1}{2} \sum_{t=0}^{T-1} \eta_t \alpha_t \Delta_t.$$

By rearranging the terms and multiplying by 2, we get

$$\sum_{t=0}^{T-1} \eta_t \alpha_t \Delta_t \leq 2 \log_+\left(\frac{T}{\delta}\right) R_0^2.$$

$\square$

## D  Proof of Theorem 1

**Theorem 1.** *Let $f : \mathbb{R}^d \to \mathbb{R}$ and suppose Assumptions 1, 2 and 4 hold. Let $\delta \in (0, 1/2)$ and let $\bar{x}$ be the output of Algorithm 1 when run for $T \geq \log_+\left(\frac{T}{\delta}\right)(64L_1R_0)^2$ steps under one of the first 3 rows of Table 1. Then with probability at least $1 - 2\delta$, the optimality gap $f(\bar{x}) - f(x^\star)$ is*

$$O\left(\frac{\log_+\left(\frac{T}{\delta}\right)\left(L_0 R_0^2 + \sqrt{\log\left(\frac{T}{\delta}\right)}\sigma R_0 \sqrt{T}\right)}{T}\right).$$

*Proof.* We begin with a comment about the light tail assumption, and then continue with the proof.

**Light-tailed noise vs. bounded noise.**  The rest of the proof is written under a modified setting: It uses Assumption 3 (bounded noise) instead of Assumption 4 (light-tailed noise), and it adjusts $\eta_t$ and $c$ by replacing instances of $3\sqrt{\log\left(\frac{T}{\delta}\right)}\sigma$ with $\sigma$. The proof obtains a bound that holds with probability at least $1 - \delta$. By Lemma 4, with probability at least $1 - \delta$, using an oracle that satisfies Assumption 4 results in the same output as using an oracle that satisfies Assumption 3 with $\sigma' = 3\sqrt{\log\left(\frac{T}{\delta}\right)}\sigma$. By using a union bound, we get that the desired bound holds under Assumption 4 with probability at least $1 - 2\delta$.

**Proof under Assumption 3.**  By Lemma 8 we have

$$\sum_{t \in \mathcal{T}_1} \eta_t \alpha_t \Delta_t \leq \sum_{t=0}^{T-1} \eta_t \alpha_t \Delta_t \leq 2 \log_+\left(\frac{T}{\delta}\right) R_0^2.$$

The criteria for belonging to $\mathcal{T}_1$, together with Lemma 7, show that this implies

$$8 \log_+\left(\frac{T}{\delta}\right)\frac{R_0^2}{T}|\mathcal{T}_1| \leq 2 \log_+\left(\frac{T}{\delta}\right) R_0^2.$$

From this, it follows that $|\mathcal{T}_1| \leq \frac{T}{2}$, and therefore $|\mathcal{T}_2| \geq \frac{T}{2}$.

Similarly, by Lemma 8 we have

$$\sum_{t \in \mathcal{T}_2} \eta_t \alpha_t \Delta_t \leq \sum_{t=0}^{T-1} \eta_t \alpha_t \Delta_t \leq 2 \log_+\left(\frac{T}{\delta}\right) R_0^2.$$

The criteria for belonging to $\mathcal{T}_2$, together with Lemma 7, show that this implies

$$\frac{1}{16}\left(11L_0 + \frac{\sigma\sqrt{T}}{R_0}\right)^{-1}\sum_{t\in\mathcal{T}_2}\Delta_t \leq 2\log_+\left(\tfrac{T}{\delta}\right)R_0^2.$$

From this, it follows that

$$\frac{1}{|\mathcal{T}_2|}\sum_{t\in\mathcal{T}_2}\Delta_t \leq \frac{32\log_+\left(\tfrac{T}{\delta}\right)R_0^2\left(11L_0 + \frac{\sigma\sqrt{T}}{R_0}\right)}{|\mathcal{T}_2|} \leq \frac{64\log_+\left(\tfrac{T}{\delta}\right)\left(11L_0R_0^2 + \sigma R_0\sqrt{T}\right)}{T},$$

where the last inequality uses the lower limit on $|\mathcal{T}_2|$.

From the definition of $\Delta_t$, and by Jensen's inequality, we obtain

$$f\left(\frac{1}{|\mathcal{T}_2|}\sum_{t\in\mathcal{T}_2}x_t\right) - f(x^\star) \leq \frac{64\log_+\left(\tfrac{T}{\delta}\right)\left(11L_0R_0^2 + \sigma R_0\sqrt{T}\right)}{T}.$$

Recall from Algorithm 1 that $|\mathcal{T}_2| \geq \frac{T}{2}$ implies $\bar{x} = \frac{1}{|\mathcal{T}_2|}\sum_{t\in\mathcal{T}_2}x_t$. Therefore the proof is complete. $\quad\square$

# E  Lemmas for proving Theorem 2

**Lemma 9.** *Let $f : \mathbb{R}^d \to \mathbb{R}$ and suppose Assumptions 1 to 3 hold. Assume Algorithm 1 is run with step size $\eta_t = \frac{R}{\sqrt{\sum_{i=0}^t \alpha_i^2\|g_i\|^2}}$ and clipping rule $\alpha_t = \min\left\{1, \frac{c}{\|g_t^c\|}\right\}$. Then for any threshold $c$,*

$$\sum_{t=0}^{T-1}\alpha_t\langle g_t, x_t - x^\star\rangle \leq 2R\sqrt{\sum_{i=0}^{T-1}\alpha_i^2\|g_i\|^2}.$$

*Proof.* Since $1/\eta_t$ is non-decreasing in $t$, a standard analysis shows that ''

$$
\begin{aligned}
\sum_{t=0}^{T-1}\alpha_t\langle g_t, x_t - x^\star\rangle &\leq \sum_{t=0}^{T-1}\frac{R_t^2 - R_{t+1}^2}{2\eta_t} + \frac{1}{2}\sum_{t=0}^{T-1}\eta_t\alpha_t^2\|g_t\|^2 \\
&\leq \frac{R_0^2}{2\eta_0} + \frac{1}{2}\sum_{t=1}^{T-1}R_t^2\left(\frac{1}{\eta_t} - \frac{1}{\eta_{t-1}}\right) + \frac{1}{2}\sum_{t=0}^{T-1}\eta_t\alpha_t^2\|g_t\|^2 \\
&\leq \frac{R^2}{2\eta_0} + \frac{R^2}{2}\sum_{t=1}^{T-1}\left(\frac{1}{\eta_t} - \frac{1}{\eta_{t-1}}\right) + \frac{1}{2}\sum_{t=0}^{T-1}\eta_t\alpha_t^2\|g_t\|^2 \\
&\leq \frac{R^2}{2\eta_{T-1}} + \frac{1}{2}\sum_{t=0}^{T-1}\eta_t\alpha_t^2\|g_t\|^2.
\end{aligned}
$$

Define $S_t := \sum_{i=0}^t \alpha_i^2\|g_i\|^2$ and $S_{-1} = 0$. Observe that $\eta_t = \frac{R}{\sqrt{S_t}}$. Then we have

$$
\begin{aligned}
\frac{1}{2}\sum_{t=0}^{T-1}\eta_t\alpha_t^2\|g_t\|^2 = \frac{1}{2}\sum_{t=0}^{T-1}\eta_t(S_t - S_{t-1}) &= \frac{R}{2}\sum_{i=0}^{T-1}\frac{S_t - S_{t-1}}{\sqrt{S_t}} \\
&= \frac{R}{2}\sum_{i=0}^{T-1}\left(\sqrt{S_t} - \sqrt{S_{t-1}}\right)\frac{\sqrt{S_t} + \sqrt{S_{t-1}}}{\sqrt{S_t}} \\
&\leq \frac{R}{2}\sum_{i=0}^{T-1}2\left(\sqrt{S_t} - \sqrt{S_{t-1}}\right) = R\sqrt{S_{T-1}}.
\end{aligned}
$$

Therefore,

$$\sum_{t=0}^{T-1} \alpha_t \langle g_t, x_t - x^\star \rangle \leq 2R \sqrt{\sum_{i=0}^{T-1} \alpha_i^2 \|g_i\|^2}.$$

$\square$

**Lemma 10.** *Let* $f : \mathbb{R}^d \to \mathbb{R}$ *and suppose Assumptions 1 to 3 hold. Additionally, assume* $6\sigma \leq c$ *and* $c \leq \|g_t^c\|$. *Then for any* $g \in \mathcal{G}(x_t)$,

$$\alpha_t^2 \|g\|^2 \leq 4(L_0 + cL_1)\alpha_t \Delta_t.$$

*Proof.* Our assumptions imply $6\sigma \leq \|g_t^c\|$, which in turn implies $\sigma \leq (1/5)\|\nabla f(x_t)\|$: otherwise,

$$\|g_t^c\| \leq \|g_t^c - \nabla f(x_t)\| + \|\nabla f(x_t)\| < \sigma + 5\sigma = 6\sigma \leq \|g_t^c\|.$$

Therefore, every oracle query to $x_t$ satisfies

$$\|\nabla f(x_t) - \mathcal{G}(x_t)\| \leq \sigma \leq (1/5)\|\nabla f(x_t)\|. \tag{4}$$

By the triangle inequalities, we obtain

$$(4/5)\|\nabla f(x_t)\| \leq \|\mathcal{G}(x_t)\| \leq (6/5)\|\nabla f(x_t)\|. \tag{5}$$

Therefore,

$$\alpha_t^2 \|g_t\|^2 \overset{(5)}{\leq} (6/5)^2 \alpha_t^2 \|\nabla f(x_t)\|^2 \tag{6}$$

$$\overset{(i)}{\leq} (6/5)^2 \alpha_t^2 \cdot 2(L_0 + \|\nabla f(x_t)\|L_1)\Delta_t$$

$$\overset{(5)}{\leq} \frac{(6/5)^2}{(4/5)} \alpha_t^2 \cdot 2(L_0 + \|g_t^c\|L_1)\Delta_t$$

$$\leq 4(\alpha_t L_0 + \alpha_t \|g_t^c\|L_1)\alpha_t \Delta_t \overset{(ii)}{\leq} 4(L_0 + cL_1)\alpha_t \Delta_t,$$

where $(i)$ uses Lemma 1 and $(ii)$ uses the definition of $\alpha_t$. $\square$

**Lemma 11.** *Assume the setting of Lemma 10, and assume that the threshold $c$ is no less than its value under "conservative + adaptive clipping". Additionally, assume that* $\log_+\left(\frac{1}{\delta}\right)(15L_1R)^2 \leq T$. *Then*

$$\alpha_t \Delta_t \geq \frac{1}{4}\frac{c}{L_1} \geq \frac{80}{T}\left(15L_0R^2 + \log_+\left(\frac{1}{\delta}\right)\sigma R\sqrt{T}\right).$$

*Proof.* By Lemma 10 we have that Equation (5) applies. Therefore,

$$\|\nabla f(x_t)\| \geq \frac{\|g_t^c\|}{6/5} \geq \frac{c}{6/5} \geq \frac{8L_0}{L_1},$$

and together with Lemma 2 we get $\|\nabla f(x_t)\| \leq 3L_1\Delta_t$. This leads to

$$\alpha_t \Delta_t = \frac{c}{\|g_t^c\|}\Delta_t \geq \frac{1}{6/5}\frac{c}{\|\nabla f(x_t)\|}\Delta_t \geq \frac{1}{6/5}\frac{c}{3L_1\Delta_t}\Delta_t \geq \frac{1}{4}\frac{c}{L_1}.$$

By our assumptions on $c$, we have

$$\frac{1}{4}\frac{c}{L_1} \geq 15\sqrt{\log_+\left(\frac{1}{\delta}\right)}\frac{R}{\sqrt{T}}c$$

$$\geq 15\sqrt{\log_+\left(\frac{1}{\delta}\right)}\frac{R}{\sqrt{T}} \cdot 15\sqrt{\log_+\left(\frac{1}{\delta}\right)}\frac{R}{\sqrt{T}}\max\left\{10L_0, \frac{\sqrt{T}}{R}\sigma\right\}$$

$$= 256\log_+\left(\frac{1}{\delta}\right)\frac{R^2}{T}\max\left\{10L_0, \frac{\sqrt{T}}{R}\sigma\right\}$$

$$\geq 125\log_+\left(\frac{1}{\delta}\right)\frac{1}{T}\left(10L_0R^2 + \sigma R\sqrt{T}\right)$$

$$\geq \frac{80}{T}\left(15L_0R^2 + \log_+\left(\frac{1}{\delta}\right)\sigma R\sqrt{T}\right),$$

where the last two inequalities are purposefully loose for later convenience. $\square$

**Lemma 12.** *Assume the setting of Lemma 9. Then for any $\delta \in (0,1)$, with probability at least $1 - \delta$,*

$$\sum_{t=0}^{T-1} \alpha_t \Delta_t \leq 32(L_0 + cL_1)R^2 + 9\log_+\left(\tfrac{1}{\delta}\right)\sigma R\sqrt{T}.$$

*If we also assume the setting of Lemma 10, then*

$$\sum_{t=0}^{T-1} \alpha_t \Delta_t \leq 25\Big(15L_0 R^2 + \log_+\left(\tfrac{1}{\delta}\right)\sigma R\sqrt{T}\Big).$$

*Proof.* The proof first uses a martingale concentration inequality and Lemma 9 to obtain a high-probability bound, and then continues to bound the stochastic gradient norms.

Define $Z_t := \alpha_t \langle \nabla f(x_t) - g_t, x_t - x^\star \rangle$. Recall from Table 1 that $\alpha_t$ is deterministic given $g_t^c$. Then the sequence $Z_0, ..., Z_{T-1}$ is a *martingale difference sequence* with respect to $\xi_{i-1} = (x_i, g_i^c)$:

$$\mathbb{E}[Z_t \mid x_t, g_t^c] = \alpha_t \langle \nabla f(x_t) - \mathbb{E}[g_t \mid x_t, g_t^c], x_t - x^\star \rangle = 0.$$

Each $Z_t$ satisfies

$$|Z_t| \leq \alpha_t \|\nabla f(x_t) - g_t\| \|x_t - x^\star\| \leq \alpha_t \sigma R_t \leq \sigma R_t \leq \sigma R.$$

Therefore, by applying Lemma 3 with $\lambda' := \left(4\sigma R\sqrt{T}\right)^{-1}$ we have that for any $\delta \in (0,1)$, with probability at least $1 - \delta$,

$$\sum_{s=0}^{T-1} \alpha_s \langle \nabla f(x_s) - g_s, x_s - x^\star \rangle \leq \frac{3}{4}\lambda' \sum_{s=0}^{T-1} \sigma^2 R^2 + \frac{1}{\lambda'}\log\left(\tfrac{1}{\delta}\right)$$

$$\leq \frac{3}{4}\lambda' T\sigma^2 R^2 + \frac{1}{\lambda'}\log\left(\tfrac{1}{\delta}\right)$$

$$\leq \frac{3}{16}\sigma R\sqrt{T} + 4\log\left(\tfrac{1}{\delta}\right)\sigma R\sqrt{T}.$$

Recall that Lemma 9 bounds $\sum_{t=0}^{T-1} \alpha_t \langle g_t, x_t - x^\star \rangle$. By summing that bound with our inequality, and by applying the gradient inequality, we get

$$\sum_{s=0}^{T-1} \alpha_s \Delta_s \leq \sum_{s=0}^{T-1} \alpha_s \langle \nabla f(x_s), x_s - x^\star \rangle$$

$$\leq \frac{3}{16}\sigma R\sqrt{T} + 4\log\left(\tfrac{1}{\delta}\right)\sigma R\sqrt{T} + 2R\sqrt{\sum_{i=0}^{T-1} \alpha_i^2 \|g_i\|^2}.$$

On iterations where $\sigma \geq (1/5)\|\nabla f(x_t)\|$, we have

$$\alpha_t^2 \|g_t\|^2 \leq \|g_t\|^2 \leq (\|g_t - \nabla f(x_t)\| + \|\nabla f(x_t)\|)^2 \leq (\sigma + 5\sigma)^2 = (6\sigma)^2.$$

On iterations where $\sigma < (1/5)\|\nabla f(x_t)\|$, we can repeat the arguments from Equations (4) to (6) (note that they do not require the additional assumptions present in Lemma 10), and thus obtain

$$\alpha_t^2 \|g_t\|^2 \leq 4(L_0 + cL_1)\alpha_t \Delta_t.$$

The two cases imply that every iteration satisfies $\alpha_t^2 \|g_t\|^2 \leq 4(L_0 + cL_1)\alpha_t \Delta_t + (6\sigma)^2$. Therefore,

$$\sum_{t=0}^{T-1} \alpha_t \Delta_t \leq \frac{3}{16}\sigma R\sqrt{T} + 4\log\left(\tfrac{1}{\delta}\right)\sigma R\sqrt{T} + 2R\sqrt{\sum_{i=0}^{T-1}\Big(4(L_0 + cL_1)\alpha_t \Delta_t + (6\sigma)^2\Big)}$$

$$= \frac{3}{16}\sigma R\sqrt{T} + 4\log\left(\tfrac{1}{\delta}\right)\sigma R\sqrt{T} + 2R\sqrt{4(L_0 + cL_1)\sum_{i=0}^{T-1}\alpha_t \Delta_t + (6\sigma)^2 T}$$

The rest of the proof is pure algebra, which we show separately in the subsequent lemma (Lemma 13).

$\square$

**Lemma 13.** *If*

$$\sum_{t=0}^{T-1} \alpha_t \Delta_t \le \frac{3}{16}\sigma R\sqrt{T} + 4\log\left(\tfrac{1}{\delta}\right)\sigma R\sqrt{T} + 2R\sqrt{4(L_0 + cL_1)\sum_{i=0}^{T-1}\alpha_t\Delta_t + (6\sigma)^2 T},$$

*then*

$$\sum_{t=0}^{T-1} \alpha_t \Delta_t \le 25\Big(15L_0 R^2 + \log_+\left(\tfrac{1}{\delta}\right)\sigma R\sqrt{T}\Big).$$

*Proof.* Denote $A := \sum_{t=0}^{T-1}\alpha_t\Delta_t$. Then we have

$$A \le \frac{3}{16}\sigma R\sqrt{T} + 4\log\left(\tfrac{1}{\delta}\right)\sigma R\sqrt{T} + 2R\sqrt{4(L_0 + cL_1)A + (6\sigma)^2 T}$$

$$\le \sqrt{\left(\frac{3}{16}\sigma R\sqrt{T} + 4\log\left(\tfrac{1}{\delta}\right)\sigma R\sqrt{T}\right)^2} + 2R\sqrt{4(L_0 + cL_1)A + (6\sigma)^2 T}$$

$$\le \sqrt{2\left(\frac{3}{16}\sigma R\sqrt{T}\right)^2 + 2\left(4\log\left(\tfrac{1}{\delta}\right)\sigma R\sqrt{T}\right)^2} + 2R\sqrt{4(L_0 + cL_1)A + (6\sigma)^2 T}.$$

Squaring both sides leads to

$$A^2 \le \left(\sqrt{2\left(\tfrac{3}{16}\sigma R\sqrt{T}\right)^2 + 2\left(4\log\left(\tfrac{1}{\delta}\right)\sigma R\sqrt{T}\right)^2} + 2R\sqrt{4(L_0 + cL_1)A + (6\sigma)^2 T}\right)^2$$

$$\le 2\left(\left(\tfrac{3}{16}\sigma R\sqrt{T}\right)^2 + 2\left(4\log\left(\tfrac{1}{\delta}\right)\sigma R\sqrt{T}\right)^2\right) + 2 \cdot 4R^2\left(4(L_0 + cL_1)A + (6\sigma)^2 T\right)$$

$$\le \frac{9}{64}\left(\sigma R\sqrt{T}\right)^2 + 4\left(4\log\left(\tfrac{1}{\delta}\right)\sigma R\sqrt{T}\right)^2 + 32R^2(L_0 + cL_1)A + 8 \cdot 36R^2\sigma^2 T$$

$$= 32R^2(L_0 + cL_1)A + \left(289 + 64\log^2\left(\tfrac{1}{\delta}\right)\right)\left(\sigma R\sqrt{T}\right)^2.$$

This is a quadratic inequality (in $A$) of the form $A^2 \le bA + c$ ($b, c \ge 0$). In such cases, every solution is less than or equal to the largest root of $A^2 - bA - c$. Therefore,

$$A \le \tfrac{1}{2}\left(b + \sqrt{b^2 + 4c}\right) \le \tfrac{1}{2}\left(b + \sqrt{b^2} + \sqrt{4c}\right) \le b + \sqrt{c}.$$

In our context, this leads to

$$\sum_{t=0}^{T-1}\alpha_t\Delta_t \le 32(L_0 + cL_1)R^2 + \sqrt{289 + 64\log^2\left(\tfrac{1}{\delta}\right)} \cdot \sigma R\sqrt{T}$$

$$\le 32(L_0 + cL_1)R^2 + \left(17 + 8\log\left(\tfrac{1}{\delta}\right)\right) \cdot \sigma R\sqrt{T}$$

$$\le 32(L_0 + cL_1)R^2 + 9\log_+\left(\tfrac{1}{\delta}\right)\sigma R\sqrt{T}$$

From the assumption on $c$, we have

$$cL_1 R^2 \le \frac{1}{L_1}\max\left\{10L_0, \frac{\sqrt{T}}{R}\sigma\right\} \cdot L_1 R^2 \le 10L_0 R^2 + \sigma R\sqrt{T}.$$

Therefore,

$$\sum_{t=0}^{T-1}\alpha_t\Delta_t \le 32\Big(L_0 R^2 + 10L_0 R^2 + \sigma R\sqrt{T}\Big) + 9\log_+\left(\tfrac{1}{\delta}\right)\sigma R\sqrt{T}$$

$$\le 352L_0 R^2 + 25\log_+\left(\tfrac{1}{\delta}\right)\sigma R\sqrt{T}$$

$$\le 25\Big(15L_0 R^2 + \log_+\left(\tfrac{1}{\delta}\right)\sigma R\sqrt{T}\Big).$$

$\square$

# F   Proof of Theorem 2

**Theorem 2.** *Assume the setting of Theorem 1 under one of the last 2 rows of Table 1. Then with probability at least $1 - 2\delta$, the optimality gap $f(\bar{x}) - f(x^\star)$ is*

$$O\left(\frac{\log_+\left(\frac{1}{\delta}\right)\left(L_0 R^2 + \sqrt{\log\left(\frac{T}{\delta}\right)}\sigma R\sqrt{T}\right)}{T}\right).$$

*Proof.* We begin with a comment about the light tail assumption, and then continue with the proof.

**Light-tailed noise vs. bounded noise.**   The rest of the proof is written under a modified setting: It uses Assumption 3 (bounded noise) instead of Assumption 4 (light-tailed noise), and it adjusts $\eta_t$ and $c$ by replacing instances of $3\sqrt{\log\left(\frac{T}{\delta}\right)}\sigma$ with $\sigma$. The proof obtains a bound that holds with probability at least $1 - \delta$. By Lemma 4, with probability at least $1 - \delta$, using an oracle that satisfies Assumption 4 results in the same output as using an oracle that satisfies Assumption 3 with $\sigma' = 3\sqrt{\log\left(\frac{T}{\delta}\right)}\sigma$. By using a union bound, we get that the desired bound holds under Assumption 4 with probability at least $1 - 2\delta$.

**Proof under Assumption 3.**   Consider iterations that satisfy $t \in \mathcal{T}_1$, that is, iterations where $c \leq \|g_t^c\|$. By Lemma 11 we have

$$\alpha_t \Delta_t \geq \frac{80}{T}\left(15 L_0 R^2 + \log_+\left(\frac{1}{\delta}\right)\sigma R\sqrt{T}\right).$$

Therefore, together with Lemma 12, we get that with probability at least $1 - \delta$,

$$\begin{aligned}
\frac{80}{T}\left(15 L_0 R^2 + \log_+\left(\tfrac{1}{\delta}\right)\sigma R\sqrt{T}\right)|\mathcal{T}_1| &\leq \sum_{t \in \mathcal{T}_1} \alpha_t \Delta_t \\
&\leq \sum_{t=0}^{T-1} \alpha_t \Delta_t \\
&\leq 25\left(15 L_0 R^2 + \log_+\left(\tfrac{1}{\delta}\right)\sigma R\sqrt{T}\right)
\end{aligned}$$

From this it follows that $|\mathcal{T}_1| \leq \frac{T}{2}$, and therefore $|\mathcal{T}_2| \geq \frac{T}{2}$.

Similarly, by Lemma 12 we have,

$$\sum_{t \in \mathcal{T}_2} \Delta_t \leq \sum_{t=0}^{T-1} \alpha_t \Delta_t \leq 25\left(15 L_0 R^2 + \log_+\left(\tfrac{1}{\delta}\right)\sigma R\sqrt{T}\right).$$

Dividing by $|\mathcal{T}_2|$, using the bound we obtained on $|\mathcal{T}_2|$ and using Jensen's inequality, we get

$$\begin{aligned}
f\left(\frac{1}{|\mathcal{T}_2|}\sum_{t \in \mathcal{T}_2} x_t\right) - f(x^\star) \leq \frac{1}{|\mathcal{T}_2|}\sum_{t \in \mathcal{T}_2} \Delta_t &\leq \frac{25\left(15 L_0 R^2 + \log_+\left(\frac{1}{\delta}\right)\sigma R\sqrt{T}\right)}{|\mathcal{T}_2|}, \\
&\leq \frac{50\left(15 L_0 R^2 + \log_+\left(\frac{1}{\delta}\right)\sigma R\sqrt{T}\right)}{T}.
\end{aligned}$$

Recall from Algorithm 1 that $|\mathcal{T}_2| \geq \frac{T}{2}$ implies $\bar{x} := \frac{1}{|\mathcal{T}_2|}\sum_{t \in \mathcal{T}_2} x_t$. Therefore the proof is complete. $\qquad\square$

# G The potentially exponential dependence of $\|\nabla f(x)\|$ and $f(x) - f(x^\star)$ on $L_1\|x - x^\star\|$

Let $L_0, L_1 \in \mathbb{R}_+$ and let $f(x) = \frac{L_0}{L_1^2}\cosh(L_1 x)$. We will show that $f$ is $(L_0, L_1)$-smooth. The first and second derivatives are

$$f'(x) = \frac{L_0}{L_1}\sinh(L_1 x) \; ; \; f''(x) = L_0\cosh(L_1 x).$$

For any $x \in \mathbb{R}$, by basic properties of $\cosh$ and $\sinh$, it holds that

$$|f''(x)| = L_0\cosh(L_1 x) = L_0\big(e^{-L_1 x} + \sinh(L_1 x)\big) = L_0 e^{-L_1 x} + L_1 f'(x).$$

When $x \geq 0$ we have $e^{-L_1 x} \leq 1$ and $f'(x) = |f'(x)|$. Therefore,

$$|f''(x)| \leq L_0 + L_1|f'(x)|.$$

When $x < 0$ we have $e^{-L_1(-x)} \leq 1$ and $f'(-x) = |f'(x)|$. Therefore,

$$|f''(-x)| \leq L_0 + L_1|f'(x)|,$$

and since $f''$ is an even function, the same holds for $|f''(x)|$. In both cases we end up with the inequality that defines $(L_0, L_1)$-smoothness.

Let us discuss, in the context of the above $f$, quantities that appear in bounds of related work. Let $x_0 \in \mathbb{R}$. The minimizer of $f$ is $x^\star = 0$, therefore $\|x_0 - x^\star\| = |x_0|$. The gradient norm at $x = x_0$ satisfies

$$\|\nabla f(x_0)\| = \frac{L_0}{L_1}|\sinh(L_1 x_0)| = \frac{L_0}{L_1}\sinh(L_1|x_0|) \geq \frac{L_0}{2L_1}(\exp(L_1|x_0|) - 1)$$

$$= \frac{L_0}{2L_1}(\exp(L_1\|x_0 - x^\star\|) - 1).$$

Additionally, the sub-optimality at $x = x_0$ satisfies

$$f(x_0) - f(x^\star) = \frac{L_0}{L_1^2}(\cosh(L_1 x_0) - 1) = \frac{L_0}{L_1^2}(\cosh(L_1|x_0|) - 1)$$

$$\geq \frac{L_0}{2L_1^2}(\exp(L_1|x_0|) - 2) = \frac{L_0}{L_1^2}(\exp(L_1\|x_0 - x^\star\|) - 2).$$

Both quantities demonstrate an exponential dependence on $L_1\|x_0 - x^\star\|$.

# H Experiments

We present additional implementation details, results of the experiments on second dataset, and results obtained with synthetic data (not modeled as a regression task).

## H.1 Implementation details of real-data experiments

**Data and computational resources.** Our experiments use the California Housing dataset [28] and the Parkinsons Telemonitoring dataset [36], which are published under "CC0" and "CC-BY 4.0" licenses, respectively. All experiments provided in this paper were run on Google Colab (with a free account) using an NVIDIA T4 GPU.

**Data preprocessing.** The data preparation begins by obtaining the dataset $(X, y)$, where $X \in \mathbb{R}^{n \times d}$ represents $n$ samples with $d$ features per sample, and $y \in \mathbb{R}^n$ represents the targets. Missing data in numerical features is replaced with the mean value, and missing data in categorical features is replaced with the most frequent value. Numeric features, as well as the targets, are standardized to have zero mean and unit variance, and categorical features are encoded as one-hot vectors. The samples are then shuffled. Finally, a column $\vec{1} \in \mathbb{R}^{n \times 1}$ is prepended to $X$ in order to have a bias term in the regression task.

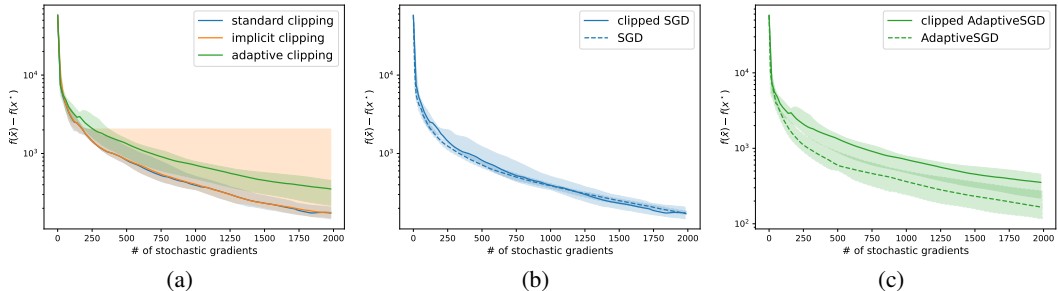

(a)                    (b)                    (c)

Figure 3: Sub-optimality of SGD variants as a function of the number of stochastic gradients used, when training a quartic loss linear regression model on the Parkinsons Telemonitoring dataset. We plot the median across 10 runs, with a shaded region showing the inter-quartile range.

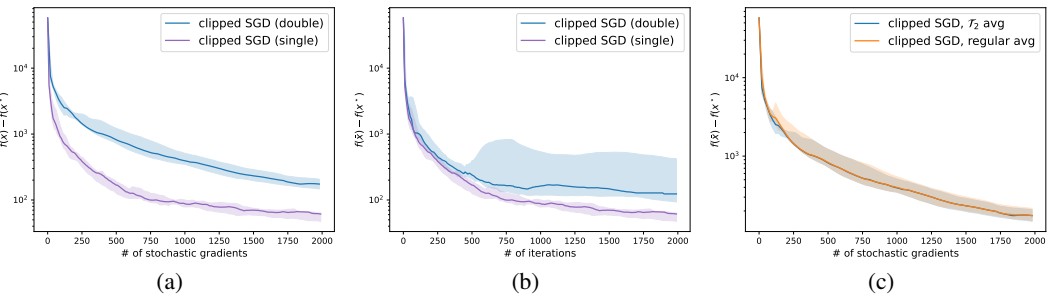

(a)                    (b)                    (c)

Figure 4: Ablations of Algorithm 1. Figures 4a and 4b compares single and double sampling by plotting sub-optimality as a function of gradient and iteration budget, respectively. Figure 4c compares different averaging methods. We plot the median across 10 runs, with a shaded region showing the inter-quartile range.

**Stepsize and clipping threshold tuning.**    We determine the clipping threshold $c$ of each method by tuning it, avoiding reliance on theoretical quantities from the definitions in Table 1. Similarly, we modify the parameter $\eta_t$ by replacing theoretical quantities with some tunable variable, which we denote as $lr$. For methods with a fixed stepsize, we simply set $\eta = lr$. For methods based on Adaptive SGD we set $\eta_t = lr \cdot (\sum_{i=0}^{t} \alpha_i^2 \|g_i\|^2)^{-1/2}$, and for "implicit clipping" we set $\eta_t = lr \cdot c/(c + \|g_t^c\|)$ (see Section 3.1 on Zhang et al. [44] for intuition).

We tune $lr$ and $c$ by performing a two-level, two-dimensional grid search. In the first-level grid, the values are geometrically spaced by a factor of 10: The values for $c$ are $(10^2, \dots, 10^7)$. The values for $lr$ are $(10^{-10}, \dots, 10^{-5})$ for SGD, $(10^{-7}, \dots, 10^{-2})$ for clipped SGD, and $(10^{-3}, \dots, 10^2)$ for both Adaptive SGD and clipped Adaptive SGD. We verify that the best candidate is never at the edge of the grid. Denoting the best candidate as $(lr^1, c^1)$, the second-level grid is defined as $\left\{(lr, c) \mid lr \in (\frac{1}{4}lr^1, \frac{1}{2}lr^1, lr^1, 2lr^1, 4lr^1), c \in (\frac{1}{4}c^1, \frac{1}{2}c^1, c^1, 2c^1, 4c^1)\right\}$.

### H.2    Additional experiments

**Parkinsons Telemonitoring dataset.**    We repeat the experiments presented in Section 4 on the Parkinsons Telemonitoring dataset [36]. The results are displayed in Figures 3 and 4 There are two notable distinctions between the results here and the results in Section 4: In Figure 3c, clipped Adaptive SGD performs worse than the others, which show similar performance. In Figure 4c, the two averaging methods seem identical.

**Synthetic data.**    We perform the same experiments, but instead of a regression-like objective, we use the function $f : \mathbb{R}^{20} \to \mathbb{R}$ given by $f(x) = \|Ax\|^4$, where $A = \text{diag}(1/20, 1/19, \dots, 1)$. We use the stochastic gradient oracle $\mathcal{G}(x) = \nabla f(x) + \xi$, where $\xi$ has iid Gaussian entries and $\sigma^2 = \text{Var}\|\xi\|^2 = 4 \cdot 10^3$. We set $T = 1000$ and $x_0 = 1.75 \cdot \vec{1}$, where $\vec{1}$ is a vector of all ones. For each tested method, we plot the median across 100 runs, with a shaded region showing the inter-quartile range. We define the stepsize of each method as $k\eta_t\alpha_t$: The values of $\alpha_t$ and $\eta_t$, unless

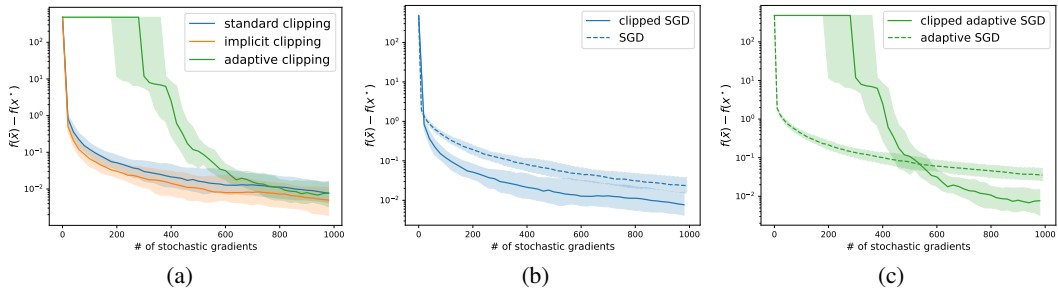

(a)                                    (b)                                    (c)

Figure 5: Sub-optimality of SGD variants as a function of the number of stochastic gradients used, on the loss $f(x) = \|Ax\|^4$ with **synthetic noise**. We plot the median across 100 runs, with a shaded region showing the inter-quartile range.

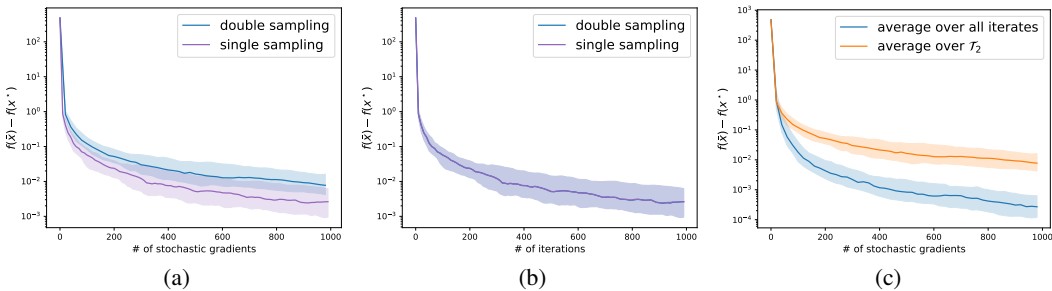

(a)                                    (b)                                    (c)

Figure 6: Ablations of Algorithm 1. Figures 6a and 6b compares single and double sampling by plotting sub-optimality as a function of gradient and iteration budget, respectively. Figure 6c compares different averaging methods. We plot the median across 100 runs, with a shaded region showing the inter-quartile range.

stated otherwise, are set according to Table 1, leveraging our knowledge of the function $f$, the time $T$ and the noise norm variance $\sigma^2$. The value $k$ is a scalar parameter that we tune: We first perform an initial grid search over $(0.01, 0.1, 1, 10, 100)$. Denoting the best value as $x$, we then perform a second grid search over $\frac{1}{4}x, \frac{1}{2}x, x, 2x, 4x$. In both searches, we verify that the optimal value is never at the edge of the grid. The results are displayed in Figures 5 and 6. Here, too, there are only a few distinctions compared to the results in Section 4: In Figure 6b, using single sampling results in the same iteration complexity as using double sampling. In Figure 6c, the difference between the two averaging methods is substantial in favor of the method from Algorithm 1.

