# OpenReview forum: "Convergence of Clipped SGD on Convex $(L_0,L_1)$-Smooth Functions"
_NeurIPS.cc/2025/Conference — NeurIPS 2025 poster_

### Official Review · Reviewer_6Ygu · 2025-06-22

**Clarity:** 3
**Significance:** 3
**Originality:** 3
**Rating:** 5
**Confidence:** 2

**Summary:**

This paper studies the high-probability convergence of a double-sampling variant of clipped SGD with five different configurations of stepsize/clipping rule/threshold under convexity and the $(L_0, L_1)$-smoothness. For this algorithm, the authors exploit a reduction from Attia and Koren to avoid some issues in previous works on the stochastic convex setting, such as the reliance on the interpolation condition or the exponential dependence of convergence rates on problem parameters. The result also recovers the best-known rate in the deterministic setting when gradient noise is zero.

**Questions:**

- How does $R$ ("$x^\*$ with distance at most $R$ from initialization $x_0$") in Assumption 1 differ from $R_0 := \\|x_0 - x^*\\|$? The authors claim that $R$ is only needed for the analysis of adaptive variants. However, the non-adaptive options in the first three lines of Table 1 still depend on $R_0$?

- Why are stepsize $\eta_t$ and threshold $c$ tuned independently in the experiments (by two-dimensional grid search), considering that they depend on the same set of problem constants? For example, when $L_0 = 0$ and $L_1=1$, $\eta_t$ in the "standard" clipped SGD should be $1/(16 c)$.

- Some previous work under convexity and the $(L_0, L_1)$-smoothness use a weaker definition of $(L_0, L_1)$-smoothness from Zhang et al. (2020a) that does not require $f$ to be twice-differentiable. Can the result in this paper be established under this weaker assumption?

Zhang, Bohang, Jikai Jin, Cong Fang, and Liwei Wang. "Improved analysis of clipping algorithms for non-convex optimization." Advances in Neural Information Processing Systems 33 (2020): 15511-15521.

- In Lemma 12, the authors first provide a bound for $\sum_t \alpha_t \Delta_t$  and then state, "If we also assume the setting of Lemma 10, then" and provide a different bound. But its proof proceeds by using results from Lemma 10 (in line 871).

- Theorem 1: $\delta\in(0,1)$ $\rightarrow$ $\delta\in(0,1/2)$?

- Lemma 13: Some of the $\sum_{t=0}^T$ should be $\sum_{t=0}^{T-1}$?

**Ethical Concerns:**

["NO or VERY MINOR ethics concerns only"]

**Final Justification:**

This paper does address certain theoretical issues in prior work on convex (L_0, L_1)-smooth optimization (such as the exponential dependence on smoothness parameters). Although they use a technique from a previous paper (not directly focused on the same problem), other parts of the proof remain non-trivial to me.

The experimental results of this paper are not extensive or large-scale, but they do not fall below the standard set by prior work on this topic.

The authors resolved most of my concerns. Although they claimed the theory can also be established under a weaker version of $(L_0,L_1)$-smoothness, they did not provide the details.

**Limitations:**

Yes

**Quality:**

3

**Strengths And Weaknesses:**

Strengths:
- The proposed algorithm and its analysis seem to be novel and interesting, especially from the theoretical perspective.
- Some design choices (such as double sampling and the conservative configuration) are motivated by the theoretical analysis.

Weaknesses:
- The theoretical analysis offers limited insight into the practical effectiveness of clipped SGD, as certain components (such as double sampling) are introduced primarily for analytical convenience.
- The empirical performance of the proposed algorithm might be sensitive to the choices of stepsize and threshold, which depend on unknown problem-specific constants. In the experiments, both parameters are tuned using a fine-grained two-level grid search.

---

> ### Author Rebuttal · Authors · 2025-07-30
>
> **Strengths:**
> > - The proposed algorithm and its analysis seem to be novel and interesting, especially from the theoretical perspective.
> > - Some design choices (such as double sampling and the conservative configuration) are motivated by the theoretical analysis.
>
> We thank the reviewer for their thoughtful and encouraging review. We are glad that the novelty of our ideas, as well as the theory-driven design choices, were found to be interesting.
>
> **Weaknesses:**
> > - The theoretical analysis offers limited insight into the practical effectiveness of clipped SGD...
> > - The empirical performance of the proposed algorithm might be sensitive to the choices of stepsize and threshold, which depend on unknown problem-specific constants...
>
> We indeed aim to position this paper as theoretical, focusing on the complexity of $(L_0,L_1)$-smooth optimization and how proper clipping helps to achieve it. Our main contribution is obtaining an additive dependence on $L_1$ and a non-exponential dependence on $L_1 R_0$, for both Clipped SGD and Clipped Adaptive SGD.
>
> **Questions:**
> > How does $R$ ("$x^\star$ with distance at most $R$ from initialization $x_0$") in Assumption 1 differ from $R_0 ≔ \lVert x_0 − x^\star \rVert$?
>
> The difference lies in what each symbol represents:
> - $R$ is used as part of our formal setup to describe a constrained optimization setting; together with the algorithm’s projection, it is an effective bound on the problem domain. AdaGrad-like algorithms require such settings.
> - $R_0$ represents the initial distance to the optimum. Note that $R_0$ may be smaller than $R$, so it is preferable for the bounds to depend on $R_0$ rather than $R$.
>
> > Why are stepsize $\eta_t$ and threshold $c$ tuned independently in the experiments (by two-dimensional grid search), considering that they depend on the same set of problem constants?
>
> We chose our tuning approach with standard practices in mind. Note that the approach is sufficient in order to answer the questions we introduce in section 4. Tuning $\eta_t$ and $c$ by tuning $L_0$, $L_1$, etc. is an interesting idea that we can consider for our next revision. However, note that directly using our formulation of $\eta_t$ and $c$ is still not trivial, as it depends on other theoretical quantities.
>
> > Some previous work under convexity and the $(L_0,L_1)$-smoothness use a weaker definition of $(L_0,L_1)$-smoothness from Zhang et al. (2020a) that does not require $f$ to be twice-differentiable. Can the result in this paper be established under this weaker assumption?
>
> We believe they can: the $(L_0,L_1)$-smoothness definition is used only in lemmas 1 & 2, which were proven in prior work without assuming twice-differentiability (see footnote on page 4).
>
> > In Lemma 12, the authors first provide a bound for $\Sigma_t \alpha_t \Delta_t$ and then state, "If we also assume the setting of Lemma 10, then" and provide a different bound. But its proof proceeds by using results from Lemma 10 (in line 871).
>
> The reference to lemma 10 in line 871 is an inexact justification on our side, and can be easily fixed (we can provide further explanation if you wish). In our next revision we will make this clearer.
>
> > - Theorem 1: $\delta \in (0,1) \to \delta \in (0,1/2)$?
> > - Lemma 13: Some of the $\Sigma_{t=0}^T$ should be $\Sigma_{t=0}^{T-1}$?
>
> You are correct, thank you for your careful reading! We will fix these in our next revision.

---

> ### Comment · Reviewer_6Ygu · 2025-08-01
>
> Thanks for your response. I think it might be helpful to include a table comparing the convergence rate established in this paper with those from relevant prior work, along with the dependence on constants, the assumptions, and some remarks. This could make it easier for readers to clearly understand the contributions of this work.

---

### Official Review · Reviewer_kUhz · 2025-06-30

**Clarity:** 2
**Significance:** 3
**Originality:** 2
**Rating:** 2
**Confidence:** 4

**Summary:**

This work analyzes the stochastic gradient descent algorithm for solving minimization problems under the $(L_0, L_1)$ smoothness condition. They provide a high probability convergence rate that matches the rate under the $L$-smoothness assumption up to poly-logarithmic terms.

**Questions:**

1. Why do you need $\mathcal{T}_1$ in the algorithm. I couldn't find where you used it.
2. Where do you use assumption 3? This appears to be a very strong assumption.

**Ethical Concerns:**

["NO or VERY MINOR ethics concerns only"]

**Final Justification:**

Based on the experimental results, the advantages of using clipped SGD are not clear to me, and the algorithm requires the knowledge of quantities that are known a priori. Moreover, I believe that, given the existence of similar results in the deterministic setting, deriving the results in the paper under the assumptions on the stochastic gradients is not particularly challenging.

**Limitations:**

Yes

**Quality:**

2

**Strengths And Weaknesses:**

Strength:

1. Analysis of stochastic gradient descent under $(L_0, L_1)$-smoothness assumption.
2. Convergence rate of SGD under $(L_0, L_1)$-smoothness assumption matches that of $L$-smoothness assumption upto a polylogarithmic factor.

Weakness:
1. In line 7 of the algorithm, the authors use projection onto the Ball of radius $R$ for adaptive methods. The value of $R$ depends on $\|x_0 - x_*\|$, which basically means you need the knowledge of $x_*$. This makes the algorithm impractical as the knowledge of $x_*$ is not known a priori.
2. All the algorithms require the knowledge of the total number of iterations $T$ to set the step sizes. This is not ideal in practice.
3.  The algorithms also need the knowledge of $\sigma$ or the variance of the gradient estimator to set the step size. This won't be available in practice.
4. In contribution point 4, the author mentions the advantage of clipped SGD over SGD in numerical experiments. However, I do not observe any advantage of clipping in the experiments.
5. The experiments are very basic. There is no proper evidence to show the advantage of gradient clipping or average over only $\mathcal{T}_2$.

---

> ### Author Rebuttal · Authors · 2025-07-30
>
> **Weaknesses:**
> > 1. In line 7 of the algorithm, the authors use projection onto the Ball of radius $R$ for adaptive methods. The value of $R$ depends on $|x_0 - x_*|$, which basically means you need the knowledge of $x_*$. This makes the algorithm impractical as the knowledge of $x_*$ is not known a priori.
>
> We do not need to know the optimum itself, but rather a bound on the initial distance from it.
>
> > 2. All the algorithms require the knowledge of the total number of iterations $T$ to set the step sizes. This is not ideal in practice.
> > 3. The algorithms also need the knowledge of $\sigma$ or the variance of the gradient estimator to set the step size. This won't be available in practice.
>
> We indeed aim to position this paper as theoretical, focusing on the complexity of $(L_0,L_1)$-smooth optimization and how proper clipping helps to achieve it. Our main contribution is obtaining an additive dependence on $L_1$ and a non-exponential dependence on $L_1 R_0$, for both Clipped SGD and Clipped Adaptive SGD.
>
> In general, it is very standard in optimization research to set the stepsize using $R_0$, $T$ and $\sigma$, and results of this sort are considered of value. Avoiding them is usually tackled in dedicated future work. We consider our “conservative” variations, which are independent of $L_1$, as a stepping stone towards this.
>
> > 4. In contribution point 4, the author mentions the advantage of clipped SGD over SGD in numerical experiments. However, I do not observe any advantage of clipping in the experiments.
>
> The synthetic experiments in the appendix show an advantage, but we agree that it is not clear how much the advantage persists in real-world losses. Thank you for noticing this. We will correct this point in our next revision.
>
> **Questions:**
> > 1. Why do you need $\mathcal{T}_1$ in the algorithm. I couldn't find where you used it.
>
> You are right in your observation that it is not necessary to maintain it. The reason we denote it explicitly in the algorithm is to make the later analysis easier to follow.
>
> > 2. Where do you use assumption 3? This appears to be a very strong assumption.
>
> We conduct most of the analysis itself under assumption 3, and then use a reduction to lift our results to hold under (the weaker) assumption 4. See lines 149-153, 229-237.

---

> > ### Comment · Reviewer_kUhz · 2025-08-02
> >
> > Thank you for your response. I do not have any further questions. However, based on the experimental results, the advantages of using clipped SGD are not clear to me. Therefore, I will maintain my current score.

---

### Official Review · Reviewer_2xE3 · 2025-07-02

**Clarity:** 3
**Significance:** 3
**Originality:** 3
**Rating:** 3
**Confidence:** 4

**Summary:**

This paper investigates stochastic gradient descent (SGD) with gradient clipping on convex functions, assuming (L0, L1)-smoothness. The authors provide theoretical results demonstrating that clipped SGD achieves a convergence rate comparable to that of standard SGD under the L-smoothness assumption, albeit with additional terms and polylogarithmic factors. The analysis is extended to adaptive versions of SGD with gradient clipping, and empirical results are presented to showcase the effectiveness of clipping.

**Questions:**

1. The paper discusses gradient clipping but does not offer detailed guidance on how to select the clipping threshold, particularly in real-world settings where data and gradients may vary. Could the authors provide more practical advice on how to tune the threshold in practice?

2. Are the assumptions (particularly the distance bound R and the Sub-Gaussian noise assumption) used in this paper standard and commonly accepted in the literature on related topics?

**Ethical Concerns:**

["NO or VERY MINOR ethics concerns only"]

**Final Justification:**

I have carefully read the author's rebuttal and the other reviewers' reviews, including the responses from other reviewers. As other reviewers have also mentioned, the experiments in this paper could be further refined to highlight the effectiveness and advantages of the proposed method. Although the author responded that this is a theoretical paper, I still believe that more solid experimental results would make the paper more convincing

**Paper Formatting Concerns:**

There are no significant formatting issues with the document.

**Quality:**

3

**Strengths And Weaknesses:**

**Strengths:**

1. The paper makes a good theoretical contribution, providing clear convergence bounds for clipped SGD in the context of convex, stochastic (L0, L1)-smooth functions.
2. The theoretical presentation is both clear and well-organized. Key ideas are succinctly explained, and the overall structure of the paper is logical and easy to follow.
3. The authors back up their theoretical claims with empirical experiments, lending credibility to their findings and demonstrating the practical relevance of their proposed methods.

**Weaknesses:**

1. While the empirical results illustrate the method's effectiveness, additional experiments on more complex tasks and larger datasets would improve the generalizability of the conclusions.
2. The double-sampling technique, while theoretically sound, seems to provide limited practical benefits. Additionally, the paper introduces several variations of SGD with clipping (standard, implicit, and adaptive), but it is not immediately clear which method performs best under different real-world conditions or how to select the most efficient one in practice.

**Typo:** Line 152 contains two semicolons.

---

> ### Author Rebuttal · Authors · 2025-07-30
>
> **Strengths:**
> > 1. The paper makes a good theoretical contribution, providing clear convergence bounds for clipped SGD in the context of convex, stochastic (L0, L1)-smooth functions.
> > 2. The theoretical presentation is both clear and well-organized. Key ideas are succinctly explained, and the overall structure of the paper is logical and easy to follow.
>
> We thank the reviewer for their thoughtful and encouraging review. We are glad that the contributions, proofs, and presentation were found to be clear and compelling.
>
> **Weaknesses:**
> > 1. ... additional experiments on more complex tasks and larger datasets would improve the generalizability of the conclusions.
> > 2. The double-sampling technique, while theoretically sound, seems to provide limited practical benefits. Additionally, the paper introduces several variations of SGD with clipping (standard, implicit, and adaptive), but it is not immediately clear which method performs best under different real-world conditions or how to select the most efficient one in practice.
>
> We indeed aim to position this paper as theoretical, focusing on the complexity of $(L_0,L_1)$-smooth optimization and how proper clipping helps to achieve it.
>
> On the matter of the several variations, please note the following.
> - In analyzing an AdaGrad-like stepsize, we provide support for the common practice of using Adam with clipping.
> - Our experiments show that all variations have similar performance, which is a useful observation in itself.
>
> > **Typo:** Line 152 contains two semicolons.
>
> Thank you for the careful reading! We will fix this typo in the next revision.
>
> **Questions:**
> > 1. The paper discusses gradient clipping but does not offer detailed guidance on how to select the clipping threshold, particularly in real-world settings where data and gradients may vary. Could the authors provide more practical advice on how to tune the threshold in practice?
>
> The most straightforward approach is to employ standard hyperparameter tuning (grid search) to tune the threshold, which for some reason is rarely performed. An observation which might prove useful is that our thresholds either scale like $\sqrt{T}$ (“standard” clipping) or remain constant (“conservative” clipping for sufficiently large $T$).
>
> > 2. Are the assumptions (particularly the distance bound R and the Sub-Gaussian noise assumption) used in this paper standard and commonly accepted in the literature on related topics?
>
> The distance bound $R$ is standard in analyses of AdaGrad and its variants. The noise assumption is also standard when proving high-probability bounds and appears in published work. In the context of $(L_0,L_1)$-smooth optimization, see [37], [42] and [43]; the last two the even stronger bounded noise assumption.

---

> > ### Comment · Reviewer_2xE3 · 2025-08-05
> >
> > Sorry for the late response. I have carefully read the author's rebuttal and the other reviewers' reviews, including the responses from other reviewers. As other reviewers have also mentioned, the experiments in this paper could be further refined to highlight the effectiveness and advantages of the proposed method. Although the author responded that this is a theoretical paper, I still believe that more solid experimental results would make the paper more convincing.

---

### Official Review · Reviewer_cvF4 · 2025-07-02

**Clarity:** 2
**Significance:** 1
**Originality:** 1
**Rating:** 2
**Confidence:** 4

**Summary:**

The authors propose a Clipped SGD with double sampling. They consider different step selection strategies and also propose an analysis of an adaptive variant of the algorithm.

**Questions:**

- On line 115, the authors claim to provide the first convergence results for stochastic convex optimization with generalized smoothness, but this is not correct. There are a number of papers considering a stochastic formulation of the problem.

- It follows from the problem statement that the authors consider a deterministic optimization problem (see line 133). Can the authors explain where the stochasticity comes from?

- Line 141 suggests that the oracle is stochastic, but due to the lack of a definition of the oracle, it is not obvious that the oracle is stochastic. The presence of stochasticity should be explicitly stated, e.g., $\mathcal{G}(x,\xi)$ or otherwise.

- Assumption 2 contains a very restricted assumption of double differentiability. Is it possible to use a relaxation of the assumption, e.g., see https://proceedings.mlr.press/v202/koloskova23a/koloskova23a.pdf? Assumption 1.2 of https://proceedings.mlr.press/v202/koloskova23a/koloskova23a.pdf?

- The motivation for the light noise is not clear as is the nature of the noise itself from Section 3. It seems to be of more interest to the ML community if the authors consider the heavy tails assumption. Will there be a difference with work https://arxiv.org/pdf/2505.20817?

- Could the authors please clarify in what sense the method can be considered adaptive, given that the threshold parameter $c$ still appears to depend on the constants $L_0$ and $R$? Additionally, could you elaborate on why the constant $L_1$ disappears from the final bounds? A more detailed explanation would be greatly appreciated.

**Ethical Concerns:**

["NO or VERY MINOR ethics concerns only"]

**Final Justification:**

I still have some concerns regarding the current version of the manuscript, which I outline below with the aim of providing constructive feedback.

Motivation of the work. While the authors provided brief responses to each of the points raised, I believe that incorporating these statements into the introduction, as currently formulated, may not significantly improve the clarity or strength of the motivation. In fact, it may risk confusing readers rather than helping them.

Clarity of Claims. I remain concerned about the claim that the desired accuracy levels of $\varepsilon_1 = 10^{-5}$ and $\varepsilon = 10^{-5555}$ can be achieved in a constant number of iterations, independent of the accuracy. This assertion appears overly strong and may require further justification or clarification.

Comparison with Relevant Prior Work. The absence of a clear discussion and comparison with related work makes it difficult to fully appreciate the contributions and significance of the results. In my view, engaging with relevant literature is essential to properly situate the work within the broader research context.

Experiments. While I appreciate the authors' perspective, I feel that the concerns raised regarding the experimental section have not been fully addressed. In particular, a comparison with related methods would significantly strengthen the empirical part of the paper.

**Quality:**

2

**Strengths And Weaknesses:**

Among the strengths of the paper, I would like to highlight that the authors provide high-probability convergence guarantees (with probability at least $1 - \delta$) for ClipSGD under $\sigma$-sub-Gaussian gradient noise. The paper also considers varying step sizes (and consequently, varying clipping rules), which adds flexibility to the analysis. Furthermore, the authors claim to establish high-probability convergence results for an adaptive version of ClipSGD. These contributions are valuable and demonstrate a serious effort on the part of the authors.

That said, I believe that in its current form, the paper is not ready for acceptance at the conference. Below I explain my point of view; if the authors believe I am mistaken, I would be glad to discuss and reconsider.

1) **Motivation of the work**: The introduction is, unfortunately, very poorly structured. In particular, the entire section preceding the statement of contributions reads more like an extended related work discussion on generalized smoothness. As a result, the paper fails to provide sufficient motivation for the problem it addresses. For instance, what is the practical or theoretical motivation for focusing on light-tailed noise and high-probability convergence? Why is double sampling used, and what is the intuition behind this choice? These and other important aspects remain unclear. I am concerned that for readers unfamiliar with the relatively recent field of generalized smoothness (and I believe there are many such readers), the paper may come across as overly theoretical and disconnected from practical relevance.

2) **Clarity of Claims**: While the technical tools employed in the paper are solid, most of the results appear to follow from previously known techniques. As such, the contributions may be viewed as incremental, and it is important that the authors carefully position their results in relation to existing literature. On this note, I would like to respectfully raise a concern regarding the accuracy of certain claims. In particular, the authors state (lines 117–118) that their results in the case $\sigma = 0$ are consistent with state-of-the-art results for the deterministic regime. However, this statement is inaccurate. For instance, the recent work https://arxiv.org/pdf/2412.17050 offers significantly stronger guarantees in the convex setting. This is especially evident in the regime where $L_0 = 0$: while the current paper suggests convergence only to a limiting neighborhood (an asymptote), the aforementioned work shows linear convergence to the exact solution. Moreover, at the end of page 3, the authors claim that linear convergence is not observed asymptotically. This statement, in the context of $L_0 = 0$, is misleading. The authors of https://arxiv.org/pdf/2412.17050 clearly highlight that when $L_0 = 0$, convergence to arbitrary precision is not only achievable but linear—a significantly stronger result than what is presented here. Additionally, Assumption 2 (as well as Definition 1) in the current paper explicitly allows for the regime $L_0 = 0$ or $L_0 \to 0$, yet this important setting is not adequately discussed. It would greatly strengthen the paper if the authors acknowledged the existing literature, accurately compared the results, and clarified their contributions in relation to prior work—particularly the work cited above, which appears to provide stronger guarantees under similar assumptions in the case $\sigma = 0$.

3) **Comparison with Relevant Prior Work**: While the paper provides a fairly detailed overview of existing work in the area of generalized smoothness, it unfortunately lacks direct comparisons between the results obtained in this paper and those in prior literature. This absence significantly weakens the clarity and contextual positioning of the contributions. To improve this, I strongly encourage the authors to include explicit comparisons with relevant previous results throughout the paper. For example, after each main theorem or corollary, it would be helpful to add a short paragraph-discussion. Such discussion would not only help clarify the novelty of the work but also make it more accessible and informative for readers less familiar with the recent literature on this topic. In addition, I strongly recommend citing and discussing the following recent papers, which are highly relevant to the topic:
https://arxiv.org/pdf/2502.00753 and https://arxiv.org/pdf/2505.20817.

4) **Experiments**: The current experimental section does not meet the expectations for a machine learning conference. The empirical results are limited to a synthetic example, with no experiments on real-world datasets or practical tasks where the proposed methods could demonstrate their relevance. Moreover, there are no comparisons with prior or related work, which makes it difficult to assess the empirical advantages or limitations of the proposed approach. In addition, the design and clarity of the figures could be substantially improved. At present, the visualizations do not effectively communicate the key insights or support the theoretical claims made in the paper. To enhance the impact of the work, I would strongly encourage the authors to: Include comparisons with at least a few baselines or state-of-the-art methods, Consider evaluating on real datasets or realistic scenarios, and Improve the figure quality (e.g., readability, overall visual design). A more thorough and polished experimental section would significantly strengthen the paper and help readers better understand the practical value of the proposed contributions.

---

> ### Author Rebuttal · Authors · 2025-07-30
>
> **Motivation of the work**:
> > The introduction is, unfortunately, very poorly structured... As a result, the paper fails to provide sufficient motivation for the problem it addresses.
>
> We respectfully disagree with this assessment, and it seems that no other reviewer shares this opinion.
> - We begin by providing motivation for studying gradient clipping, introducing the concept of $(L_0,L_1)$-smoothness, and highlighting its significance in the context of gradient clipping.
> - We then briefly mention an interesting pattern observed in non-convex analyses (the low-order dependence on $L_1$).
> - We move on to discuss the motivation and challenge of analysing the stochastic convex setting.
> - Then we list our contribution and briefly describe the double-sampling technique.
>
> > For instance, what is the practical or theoretical motivation for focusing on light-tailed noise and high-probability convergence? Why is double sampling used, and what is the intuition behind this choice? These and other important aspects remain unclear.
>
> The choices you mention are not rooted in an underlying motivation, but rather as a means to realize our goal, which is understanding the theoretical complexity of $(L_0,L_1)$-smooth optimization. As such, we believe that it is more appropriate to discuss them in section 3 (lines 145-153, 160-165).
>
> Having said that, we can revise the introduction to note the following:
> - High-probability guarantees are generally favorable compared to in-expectation guarantees.
> - The choice of double-sampling is motivated by the challenge of bias in gradient clipping.
> - Assuming light tails follows published work such as [37], [42] and [43] which, even under this assumption, contribute insightful ideas and are considered valuable.
>
> **Clarity of Claims**:
> > ... most of the results appear to follow from previously known techniques. As such, the contributions may be viewed as incremental...
>
> While the general outline of the proof may appear familiar, our argument relies on a novel choice of clipping threshold that allows us to decouple the effects of gradient clipping and gradient noise (Lemma 6), as well as careful use of concentration inequalities (Lemma 8) that are essential for enabling the case-splitting argument at the core of our proof.
>
> To evaluate the novelty of these techniques we note that:
> - These techniques do not appear in analyses of clipped-SGD under generalized smoothness in the non-convex case.
> - Prior published work have already studied the convex generalized smooth setting, yet they do not manage to remove the exponential dependence on $L_1 R_0$.
>
> > ... the authors state (lines 117–118) that their results in the case $\sigma = 0$ are consistent with state-of-the-art results for the deterministic regime. However, this statement is inaccurate. For instance, the recent work [link] offers significantly stronger guarantees in the convex setting. This is especially evident in the regime where $L_0=0$: while the current paper suggests convergence only to a limiting neighborhood (an asymptote), the aforementioned work shows linear convergence to the exact solution.
>
> Could you please explain why, in the case $L_0 = \sigma = 0$, you think our paper suggests convergence only to a limiting neighborhood? When plugging these values to theorem 3, it shows the sub-optimality gap is 0. That is, we show convergence to the exact solution in a constant number of iterations, where the constant is any solution to the inequality $T \geq \log(T) (L_1 R_0)^2$.
>
> > Moreover, at the end of page 3, the authors claim that linear convergence is not observed asymptotically. This statement, in the context of $L_0 = 0$, is misleading.
>
> Indeed, that claim is stated for the general $(L_0,L_1)$-smooth case. We can alter it in our next revision to highlight the difference between the general case and $L_0=0$.
>
> **Comparison with Relevant Prior Work:**
> > ... lacks direct comparisons between the results obtained in this paper and those in prior literature...
>
> We compare our work to existing literature in lines 108-119. Let us reiterate the key point from that part: a crucial factor that affects all known bounds in the stochastic setting is the dependence on either $\Delta$ or $M$, both of which may be exponential in $L_1 R_0$. One of the main novelties in our work is surpassing this exponential dependence.
>
> > In addition, I strongly recommend citing and discussing the following recent papers, which are highly relevant to the topic: [link] and [link].
>
> We would be happy to discuss the first paper in our next revision. The second paper was released after the neurips deadline, so we think it is fair to view it as subsequent to our work.
>
> Let us elaborate on the first paper: it studies stochastic mirror descent under $\ell$-smoothness (a generalization beyond $(L_0,L_1)$-smoothness), and shows a convergence rate with an implicit exponential dependence on $L_1 R_0$. This dependency can be deduced by looking at their proof: eqn. 202 (page 62) relies on $L_{t-1}$, a function of the initial gradient norm. The authors claim that this quantity is $O(1)$, but it is only because they treat the initial gradient norm as $O(1)$.
>
> **Experiments:**
> > The current experimental section does not meet the expectations for a machine learning conference. The empirical results are limited to a synthetic example, with no experiments on real-world datasets or practical tasks where the proposed methods could demonstrate their relevance.
>
> Our main experiments are **not** synthetic; they use two real-world datasets and test the (practical) task of linear regression.
>
> > Moreover, there are no comparisons with prior or related work, which makes it difficult to assess the empirical advantages or limitations of the proposed approach.
>
> Since we tune $\eta$, prior clipping methods are essentially equivalent to single-sampling clipping, which we do test. We also compare our methods to SGD/Adaptive SGD without clipping.
>
> **Questions:**
> > - On line 115, the authors claim to provide the first convergence results for stochastic convex optimization with generalized smoothness, but this is not correct. There are a number of papers considering a stochastic formulation of the problem.
>
> Our exact claim is that we provide “the first rate… **without exponential dependence on $L_1 R_0$**”. If you believe this is incorrect, please show an example and we will make sure to correct this in our next revision.
>
> > - It follows from the problem statement that the authors consider a deterministic optimization problem (see line 133). Can the authors explain where the stochasticity comes from?
> > - Line 141 suggests that the oracle is stochastic, but due to the lack of a definition of the oracle, it is not obvious that the oracle is stochastic. The presence of stochasticity should be explicitly stated, e.g., $\mathcal{G}(x,\xi)$ or otherwise.
>
> Stochasticity comes from the gradient oracle (as explicitly stated). Our notation is consistent with work such as Zhang et al. (2019).
>
> > - Assumption 2 contains a very restricted assumption of double differentiability. Is it possible to use a relaxation of the assumption, e.g., see [link]? Assumption 1.2 of [link]?
>
> It is probably possible to relax this, as it only affects basic smoothness lemmas.
>
> > - The motivation for the light noise is not clear as is the nature of the noise itself from Section 3. It seems to be of more interest to the ML community if the authors consider the heavy tails assumption. Will there be a difference with work [link]?
>
> We leave heavy tails for future work.
>
> > - Could the authors please clarify in what sense the method can be considered adaptive, given that the threshold parameter $c$ still appears to depend on the constants $L_0$ and $R$?
>
> The use of the word “adaptive” comes from $\eta_t$ resembling Adaptive SGD (aka AdaGrad-Norm).
>
> > Additionally, could you elaborate on why the constant $L_1$ disappears from the final bounds? A more detailed explanation would be greatly appreciated.
>
> The intuition is that, roughly speaking, every time clipping occurs we decrease the distance to the minimum by $\Omega(1/L_1^2)$ (see Claim 2 in the proof sketch and think of $T$ as a fraction of $(L_1 R_0)^2$). Therefore, there can be at most $(L_1 R_0)^2$ such iterations. The rest of the trajectory behaves as though the function was $L_0$-smooth.

---

> > ### Comment · Reviewer_cvF4 · 2025-08-03
> >
> > I would like to sincerely thank the authors for their detailed and thoughtful responses. Nevertheless, I still have some concerns regarding the current version of the manuscript, which I outline below with the aim of providing constructive feedback.
> >
> > - Motivation of the work. While the authors provided brief responses to each of the points raised, I believe that incorporating these statements into the introduction, as currently formulated, may not significantly improve the clarity or strength of the motivation. In fact, it may risk confusing readers rather than helping them.
> >
> > - Clarity of Claims. I remain concerned about the claim that the desired accuracy levels of $\varepsilon_1 = 10^{-5}$ and $\varepsilon_2 = 10^{-5555}$ can be achieved in a constant number of iterations, independent of the accuracy. This assertion appears overly strong and may require further justification or clarification.
> >
> > - Comparison with Relevant Prior Work. The absence of a clear discussion and comparison with related work makes it difficult to fully appreciate the contributions and significance of the results. In my view, engaging with relevant literature is essential to properly situate the work within the broader research context.
> >
> > - Experiments. While I appreciate the authors' perspective, I feel that the concerns raised regarding the experimental section have not been fully addressed. In particular, a comparison with related methods would significantly strengthen the empirical part of the paper.
> >
> > >**Stochasticity comes from the gradient oracle (as explicitly stated). Our notation is consistent with work such as Zhang et al. (2019).**
> >
> > This issue remains relevant in the current version of the paper. Unfortunately, the response does not appear to significantly improve the clarity or understanding for the reader.
> >
> > >**It is probably possible to relax this, as it only affects basic smoothness lemmas.**
> >
> > Could the authors provide detailed proofs?
> >
> > >**The motivation for the light noise is not clear as is the nature of the noise itself from Section 3. It seems to be of more interest to the ML community if the authors consider the heavy tails assumption. Will there be a difference with work [link]?**
> >
> > This issue is still of concern, especially regarding the lack of comparison with work [1].
> >
> > [1] Chezhegov, S. et al. (2025). Convergence of Clipped-SGD for Convex $(L_0, L_1) $-Smooth Optimization with Heavy-Tailed Noise.

---

> > > ### Author Response · Authors · 2025-08-05
> > >
> > > We thank the reviewer for the constructive comments, and while our views on some issues remain different, we will take your suggestions under consideration.
> > >
> > > > Clarity of Claims. I remain concerned about the claim that the desired accuracy levels of $\varepsilon_1 = 10^{-5}$ and $\varepsilon_2 = 10^{-5555}$ can be achieved in a constant number of iterations, independent of the accuracy. This assertion appears overly strong and may require further justification or clarification.
> > >
> > > In the noiseless case, when $L_0=0$, every gradient step reduces the squared distance to the optimum by at least $\Omega(1/L_1^2)$, and therefore the optimum must be reached in $O((R_0 L_1)^2)$ steps. This result was known before our work, see, e.g., [11], whose complexity bound also states exact convergence within $O((R_0 L_1)^2)$ steps for $L_0=0$. That said, it is not clear to us whether meaningful examples with $L_0=0$ even exist, as single-example logistic regression has $R_0 = \infty$.
> > >
> > > > Could the authors provide detailed proofs?
> > >
> > > Let us explain why the existing proofs already hold under the relaxed version of $(L_0,L_1)$-smoothness:
> > >
> > > Lemma 1 is proved in [15] (Lemma A.2), in which twice-differentiability is not assumed ([15] themselves use a proof from [42], which at first appears to assume twice-differentiability, but Appendix A.1 in [42] shows that it isn’t the case).
> > >
> > > The rest of our analysis, throughout the whole paper and appendices, never assumes twice-differentiability nor mentions Hessians, neither implicitly nor explicitly.
> > >
> > > > This issue is still of concern, especially regarding the lack of comparison with work [1].
> > >
> > > We view [1] as subsequent to our work.

---

> > > > ### Comment · Reviewer_cvF4 · 2025-08-08
> > > >
> > > > Dear Authors,
> > > >
> > > > Thank you for your response. Unfortunately, my concerns were not fully addressed. Nevertheless, I encourage you to take the above comments into account and consider resubmitting your work to a future conference.
> > > >
> > > > Best of luck!
> > > >
> > > > Sincerely,
> > > >
> > > > Reviewer cvF4

---

### Comment · Area_Chair_KhJX · 2025-08-05

Dear Reviewers,

Thank you again for your time and efforts in reviewing papers for NeurIPS 2025.

I am writing to remind you that **active participation in the author-reviewer discussion phase is mandatory**. According to the guidelines from the NeurIPS program chairs, reviewers are **required to engage directly with the authors in the discussion thread**, especially in response to their rebuttals.

Please note the following important policy:

- Simply reading the rebuttal or internally considering it is **not sufficient** -- reviewers must **post at least one message to the authors**, even if it is only to confirm that their concerns were resolved. If they have not been addressed, please explain why.

- **Acknowledging the rebuttal without any engagement with the authors will be considered insufficient**. I am obligated to flag such cases using the *InsufficientReview* mechanism, which may **impact future reviewing invitations and result in desk rejection of your own submissions**.

If you have not yet responded to the authors in the discussion thread, I kindly ask you to do so **as soon as possible**, and **no later than August 8, 11:59pm AoE**.

Please don't hesitate to reach out to me if you have any questions or concerns.

Best regards,

AC

---

### Decision · Program_Chairs · 2025-09-17

**Decision:**

Accept (poster)

**Comment:**

# Reiew by AC

This paper addresses the stochastic minimization problem under convexity, $(L_0, L_1)$-smoothness, and a light-tailed noise assumption. The authors derive new convergence results for Clip-SGD (and its AdaGrad-type variant) applied to this setting. Their rates achieve logarithmic dependence on the failure probability and avoid exponentially large factors in $L_1 R_0$, where $R_0$ bounds $||x_0 - x_\ast||$. To the best of my knowledge, these are the first convergence guarantees - among both in-expectation and high-probability results - without exponential dependence on $L_1 R_0$, while also attaining the correct (non-accelerated) dependence on the number of iterations $T$ in the convex $(L_0, L_1)$-smooth case without requiring large batch sizes. Prior work either focused on non-convex problems or relied on large-batch regimes. Overall, I view this as a strong theoretical contribution, as deriving high-probability bounds in the $(L_0, L_1)$-smooth setting is highly non-trivial.

Weaknesses include:
- Reliance on the light-tailed noise assumption,
- Inferior empirical performance of clipping with double sampling compared to standard clipping.

However, I believe these weaknesses are outweighed by the theoretical strengths.

I would also like to emphasize the theoretical elegance of double sampling: under light-tailed noise, there is no need to use the same sample for both the update direction and the clipping coefficient (a necessity under heavy-tailed noise). Here, clipping serves primarily to handle $(L_0, L_1)$-smoothness and eliminate exponentially large dependencies on $L_1 R_0$.

# Reviewers' opinions

Three reviewers recommended rejection, while only Reviewer 6Ygu supported acceptance. Below, I explain why I believe the criticisms raised do not justify rejection.

## Reviewer cvF4

Concerns: motivation, clarity of claims, comparison with related work, and limited experiments.

- Motivation: Motivation: I find the motivation sufficient. Light-tailed noise is a standard assumption in stochastic optimization and does arise in practice (e.g., ResNet50 on ImageNet, see [Zhang, Jingzhao, et al. "Why are adaptive methods good for attention models?" Advances in Neural Information Processing Systems 33 (2020): 15383-15393]).

- Clarity of claims: This is a minor issue, easily addressed with an additional remark clarifying the better initial-stage rate.

- Comparison with related work: The comparisons are clear to experts. While they could be made more explicit, I do not see this as essential. Moreover, asking for comparisons with a paper posted on arXiv after the submission deadline is not appropriate.

- Experiments: The experiments are indeed limited, but this alone does not justify rejection given the theoretical contribution.

## Reviewer 2xE3

Concerns: experiments and double sampling - largely the same weaknesses I acknowledged. As argued above, I believe the theoretical contributions outweigh these limitations.

## Reviewer kUhz

Concerns: limited experiments, projection onto a ball of radius $R$, and the need to know $\sigma$ and $T$ for stepsize selection.

- Projection: This is required only for the adaptive method and is a standard assumption in AdaGrad-type analyses (bounded gradients or bounded domain).
- Stepsize dependence on $\sigma$ and $T$: While a limitation, it is common across many works in stochastic optimization. Even with these assumptions, the results are non-trivial and advance the theoretical understanding of stochastic methods.

## Reviewer 6Ygu

This reviewer found the proposed algorithm and results novel and theoretically interesting. Their listed weaknesses overlap with those I already noted.

# Final recommendation

This paper makes a meaningful contribution to the theory of stochastic optimization. The results are novel, the double-sampling idea is theoretically elegant, and the main technical achievement - removing exponential dependence on $L_1 R_0$ in the convex $(L_0, L_1)$-smooth setting - addresses an important gap. While experiments are limited and some assumptions could be stronger, the paper’s theoretical contributions are substantial and clearly outweigh the weaknesses.

**Therefore, despite three negative reviews, I recommend that this paper be accepted to NeurIPS.**